# Myelo-lymphoid lineage restriction occurs in the human haematopoietic stem cell compartment before lymphoid-primed multipotent progenitors

Serena Belluschi[1], Emily F. Calderbank[1], Valerio Ciaurro[1], Blanca Pijuan-Sala[1], Antonella Santoro[1], Nicole Mende[1], Evangelia Diamanti[1], Kendig Yen Chi Sham[1], Xiaonan Wang[1], Winnie W. Y. Lau [1], Wajid Jawaid [1], Berthold Göttgens[1] & Elisa Laurenti [1]

Capturing where and how multipotency is lost is crucial to understand how blood formation is controlled. Blood lineage specification is currently thought to occur downstream of multipotent haematopoietic stem cells (HSC). Here we show that, in human, the first lineage restriction events occur within the $CD19^-CD34^+CD38^-CD45RA^-CD49f^+CD90^+$ ($49f^+$) HSC compartment to generate myelo-lymphoid committed cells with no erythroid differentiation capacity. At single-cell resolution, we observe a continuous but polarised organisation of the $49f^+$ compartment, where transcriptional programmes and lineage potential progressively change along a gradient of opposing cell surface expression of CLEC9A and CD34. $CLEC9A^{hi}CD34^{lo}$ cells contain long-term repopulating multipotent HSCs with slow quiescence exit kinetics, whereas $CLEC9A^{lo}CD34^{hi}$ cells are restricted to myelo-lymphoid differentiation and display infrequent but durable repopulation capacity. We thus propose that human HSCs gradually transition to a discrete lymphoid-primed state, distinct from lymphoid-primed multipotent progenitors, representing the earliest entry point into lymphoid commitment.

[1] Department of Haematology and Wellcome and MRC Cambridge Stem Cell Institute, University of Cambridge, Cambridge, UK. These authors contributed equally: Serena Belluschi, Emily F. Calderbank  Correspondence and requests for materials should be addressed to E.L. (email: el422@cam.ac.uk)

Production of all mature blood cell types results from the concerted action of haematopoietic stem (HSC) and progenitor cells. HSCs have been historically and operationally defined as the only cells capable of producing all blood cell types for the lifetime of an individual or upon successive rounds of transplantation. Definitive evidence that multipotency and long-term blood production can coexist within a single cell was provided first in mouse[1] then in human[2]. It is generally understood that whereas cells in the HSC and multipotent progenitors (MPP) compartment are multipotent, the first reported major event of lineage restriction occurs downstream of HSCs/MPPs to separate myelo-lymphoid (My/Ly) and myelo-erythroid (My/Ery) fates. This corresponds to the separation into lymphoid-primed multipotent progenitor (LMPP)/multi lymphoid progenitor (MLP)[3–6] and common myeloid progenitor (CMP) compartments[7,8]. My-committed cells then segregate from the Ly-committed ones in one branch, and from Ery-committed cells in the other branch.

Understanding when and how multipotency is lost is crucial to capture how the haematopoietic system responds to stress and how leukaemia is initiated[9]. In the classical model, the transition from multipotent to lineage-restricted cells occurs exclusively outside of the HSC/MPP compartment. Recently, single cell in vitro differentiation experiments with progenitor cells[10–13], clonal tracking in mouse models[14,15] and extensive single-cell RNA-seq of mouse and human stem and progenitor cells[16–18] have demonstrated that within the progenitor compartment the vast majority of cells differentiate along a single lineage, instead of at least two as previously thought. Upstream, single phenotypic HSCs display heterogeneous and stereotypic cell-autonomous behaviours[19]. Notably, HSCs vary in the relative proportions of differentiated progeny that they produce[20,21]. In mice, platelet-biased, My-biased and Ly-biased HSCs have been reported[22–28]. Similarly, biased MPP subsets have also been identified[29,30]. The molecular basis of these distinct differentiation behaviours remains to be clarified. This body of work also leaves unanswered whether lineage restriction events occur exclusively in the rare multipotent cells present within the short-lived progenitor compartment or if lineage restriction events are already initiated among long-term repopulating HSCs.

In human, purification strategies based on differential expression of CD49f and CD90 enrich for long-term (49f[+]) and short-term (49f[−]) repopulating HSCs, with distinct cell cycle properties, but similar My and Ly potential[2,31]. Recent work has proposed that Ery and megakaryocytic (Meg) fates branch off directly from 49f[−] cells[12,18]. In contrast, Ly molecular priming and commitment is thought to occur just downstream of HSCs/MPPs[4–6,32]. No systematic characterisation at single-cell resolution of the lineage potential of 49f[+] HSCs and their molecular programmes has been reported to date. Here, we measure the differentiation potential towards the My, Ery, Meg and Ly lineages of more than 5500 single human HSC/MPP cells and single 49f[+] HSCs in vitro. Coupling this approach with index-sorting technology and single-cell RNA-seq, we uncover that, in contrast to the accepted model, lineage restriction events towards My/Ly fates already occur within 49f[+] HSCs. We show that within a continuous but highly structured molecular landscape, progression to a CLEC9A-[lo]CD34[hi] phenotype corresponds to the earliest transition of human HSCs to a discrete erythroid-null lymphoid-primed cell type characterised by fast quiescent exit kinetics and infrequent but durable repopulation capacity.

## Results

### Heterogeneous in vitro differentiation of single human HSCs.
To characterise the differentiation potential of single human phenotypic HSCs along the My, Ly, Meg, and Ery lineages, we optimised an MS5 stromal cells[33] based assay initially developed for measuring differentiation of human progenitors towards the My/Ery/Meg branches[12]. Our conditions support all these lineages and Natural Killer (NK) cell differentiation. We index-sorted 435 single HSC/MPP pool cells (CD19[−]CD34[+]CD38[−]CD45RA[−], Supplementary Fig. 1a) from six individual umbilical cord blood (CB) samples, recording the cell surface intensity of nine cell surface proteins (CD19, CD34, CD38, CD45RA, CD90, CD49f, CD10, CLEC9A, CD117). We included CD117, which levels have been previously shown to mark human HSCs with different repopulation capacities[34], and CLEC9A, a receptor for which the mRNA was expressed at significantly higher levels in 49f[+] HSCs (CD19[−]CD34[+]CD38[−]CD45RA[−]CD90[+]CD49f[+], on average 13.7% of the HSC/MPP pool) than in 49f[−] HSCs (CD19[−]CD34[+]CD38[−]CD45RA[−]CD90[−]CD49f[−], on average 13.4% of the HSC/MPP pool) (Supplementary Fig. 1a)[31,32]. The expression of CLEC9A, a SYK-coupled C-type lectin receptor[35], has not been reported outside of the dendritic cell compartment to date. Flow cytometry confirmed that approximately 80% of 49f[+] HSCs are CLEC9A[+] compared to 50% of 49f[−] HSCs with progenitor populations lower still (Supplementary Fig. 1b).

In this assay, around 80% of single cells reproducibly produced colonies of nine different types across individual CBs and experiments (Supplementary Fig. 1c, d and Fig. 1a). No single HSC/MPP pool cells produced quadrilineage colonies (Fig. 1a), but approximately 53% of the cells produced colonies with 2 or 3 lineages, 6.5% of which of the My/NK/Ery type (Fig. 1a). Among unilineage colonies, around 80% were My-only colonies, Ery- and NK-only colonies were similarly abundant and a very small percentage of Meg-only colonies were recorded (< 1%, Fig. 1a). My colonies contained either monocytes, granulocytes or both (Supplementary Fig. 1e).

This reproducible pattern of colony formation could be driven by pre-existing heterogeneity in lineage output within the HSC/MPP pool, potentially related to the expression of different levels of cell surface markers. In a principal component analysis (PCA) of the expression of all cell surface markers at the time of index-sort, cells that generated My-only colonies were evenly scattered across the PCA space (Supplementary Fig. 1f). However, the distribution on PC1 of cells with Ery and My/Ery colony output was significantly shifted from that of cells with My/NK and NK output (Fig. 1b, c). Cells producing My/NK/Ery colonies were located in between these groups, indicating functional polarisation in the HSC/MPP pool.

We next verified whether such polarisation exists within the 49f[+] HSC compartment, where 1 in 10 cells repopulate long-term in xenograft models[2]. We index-sorted 819 49f[+] HSCs from four independent CBs and recorded for each cell the differentiated colony output and the time of first division, because quiescence exit kinetics are differentially regulated among HSCs[31]. Similar to the HSC/MPP pool, approximately 80% of single 49f[+] HSCs formed a colony (Supplementary Fig. 1c, d). Here, less than 1% of the cells gave rise to quadrilineage colonies (Fig. 1d), approximately 10% of multilineage colonies were of the My/NK/Ery type and the only unilineage colonies produced were My-only (Fig. 1d and Supplementary Fig. 1e), indicating an overall more primitive compartment. Again, single 49f[+] HSCs generating My-only colonies were spread across the cell surface marker PCA space (Supplementary Fig. 1g), but those producing My/Ery colonies and My/NK colonies were distributed unequally along the PC1 and PC2 axes (Fig. 1e, f). Cells producing My/NK/Ery or My/NK/Ery/Meg colonies had the same distribution as those producing My/Ery colonies (Fig. 1e, f). Interestingly, single 49f[+] HSCs generating colonies containing Ery cells completed their first division significantly later than those producing colonies containing NK cells (p < 0.001, Fig. 1g and Supplementary Fig. 1h).

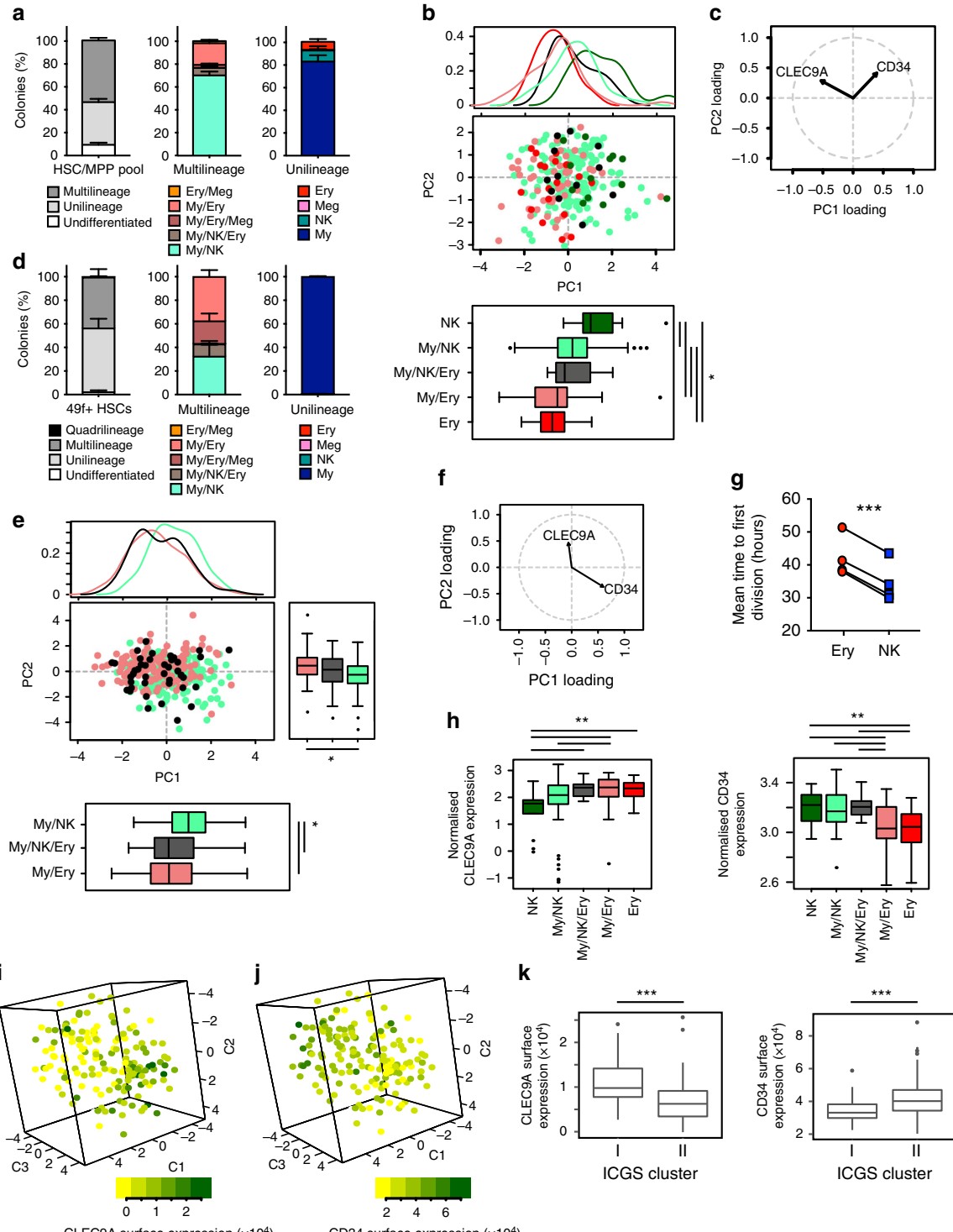

Despite the delay in the first division of cells differentiating towards the Ery lineage, My/Ery colonies were averagely significantly larger than any other type of colony, including multilineage colonies (Supplementary Fig. 1i) suggesting that, at least in this assay, expansion is maximal along the Ery differentiation branch. In summary, we uncovered pre-determined functional polarisation within the HSC/MPP pool and the highly purified 49f+ HSC compartment, which separates single cells with My/Ly potential from those with My/Ery or multilineage potential.

**Continuum of My/Ly-priming along a CD34 and CLEC9A gradient**. Analysis of PCA vector loadings showed that the cell surface expression of CD34 and CLEC9A drives most of the variance on PC1 in HSC/MPP pool (Fig. 1b, c) and on both PC1 and PC2 in 49f+ HSCs (Fig. 1e, f). Consistently, single cells that produced Ery or My/Ery colonies had significantly lower levels of CD34 and higher levels of CLEC9A cell surface expression than those producing My/NK or NK colonies (Fig. 1h and Supplementary Fig. 2a, median fluorescence intensity shifted, respectively, by 40% for CLEC9A and 13% for CD34). Interestingly,

**Fig. 1** A continuum of differentiation outputs and transcriptomes within the human HSC/MPP pool and 49f[+] HSCs compartments. **a**, **d** Percentage of colonies of the indicated type derived from HSC/MPP pool single cells ($n = 435$ colonies from six independent CBs) (**a**) and from 49f[+] HSCs ($n = 819$ colonies from four independent CBs) (**d**); mean ± SEM is shown. **b**, **c**, **e**, **f**, Principal component analysis (PCA) of the surface marker expression at the time of sort of single HSC/MPP pool cells ($n = 435$ colonies from six independent CBs) (**b**, **c**) and of 49f[+] HSCs ($n = 714$ colonies from four independent CBs) (**e**, **f**). Colours indicate the type of differentiated colony produced by each single-cell after culture. Density plots (top panel) and boxplots (bottom panel) of PC1 (**b**, **e**), and PC2 values (right panel, **e**) from single cells producing indicated types of differentiated colonies; *$p < 0.05$ by Kruskal–Wallis test with multiple comparisons. **c**, **f** Cell surface markers with the highest PC loadings are shown. **g** Mean time to first division of single cells producing colonies containing the indicated cell types (EC$_{50}$ of non-linear fit of cumulative first division kinetics); $n = 4$ independent experiments each with an independent CB (with respectively 19, 98, 34, 230 single cells per experiment), ***$p < 0.001$ by two-sided paired $t$-test. **h** Normalised cell surface expression of CLEC9A (left) and CD34 (right) at the time of sort (same HSC/MPP pool cells as in **b**) for single cells producing the indicated types of differentiated colonies. Of note, in **e** and **h**, My/NK/Ery includes My/NK/Ery and My/NK/Ery/Meg colonies. **$p < 0.01$ by Kruskal–Wallis test with multiple comparisons. **i**–**k** Single-cell RNA-seq analysis of 49f[+] HSCs ($n = 169$ single cells that passed quality control). **i**, **j** Cell surface expression of CLEC9A (**i**) and CD34 (**j**) overlaid on the tSNE representation of 49f[+] HSCs. **k** Cell surface expression levels of CLEC9A (left) and CD34 (right) in the 49f[+] HSCs of the indicated ICGS clusters; ***$p < 0.001$ by two-sided unpaired $t$-test. All boxplots show median, interquartile and 5–95 percentile

milder but significant shifts in expression of CLEC9A and CD34 cell surface expression were also observed between 49f[+] HSCs that generated My/NK colonies with a Ly-bias (number of NK cells in the colony greater than number of My cells) and those with a My-bias, as well as between cells generating My/Ery colonies with an Ery-bias (number of Ery cells in the colony greater than number of My cells) and those with a My-bias (median fluorescence intensity shifts of 4 to 25%, Supplementary Fig. 2b). These data indicate that cells with distinct differentiation outputs are distributed on a continuum of anticorrelating CLEC9A and CD34 cell surface expression.

To investigate if the observed functional polarisation of the 49f[+] compartment is mirrored at the transcriptional level, we index-sorted a total of 192 single 49f[+] HSCs and performed single-cell RNA-seq. Eighty-eight percent of single cells passed quality control. In all, 49f[+] HSCs occupy a continuous transcriptional space as observed with three distinct dimensionality reduction techniques (Fig. 1i–j and Supplementary Fig. 2c–f).

We observed that cell surface expression of CLEC9A progressively decreased from one end to the other, whereas CD34 levels gradually changed in the opposite direction to CLEC9A (Fig. 1i–j and Supplementary Fig. 2c–f). To unbiasedly identify possible clusters of cells driven by specific gene signatures, we applied the Iterative Clustering and Guide-gene Selection (ICGS) algorithm[36] and found two clusters, respectively, comprising 53 and 116 cells (Supplementary Fig. 2g). Cluster I cells expressed significantly higher levels of CLEC9A and significantly lower levels of CD34 on their surface than Cluster II cells (Fig. 1k). In summary, opposing gradients of CLEC9A and CD34 cell surface expression correlate with the observed polarisations of the transcriptional landscape and of in vitro lineage output in human HSCs.

**Prospective enrichment of My/Ly and My/Ery subsets in vitro.** Single cells with bilineage read-outs in vitro could either be multipotent HSCs with strong lineage-biases, or irreversibly lineage-restricted cells. To discriminate between these hypotheses, we sought to develop prospective purification strategies for novel subpopulations within the HSC/MPP pool and 49f[+] HSC compartments with these specific differentiation behaviours in vitro. Using the single-cell data shown in Fig. 1a–f, we systematically tested all possible gate thresholds along the CLEC9A/CD34 spectrum in silico to determine the sorting strategy that maximises enrichment of bipotent lineage output in vitro, either My/Ery or My/NK (Fig. 2a). From this analysis, we predicted the CD34[lo]CLEC9A[hi] fractions (hereafter called Subset1 and 49f[+] Subset1) and the CD34[hi]CLEC9A[lo] fractions (hereafter termed Subset2 and 49f[+] Subset2) from HSC/MPP or 49f[+] HSC pools to be, respectively, enriched for single cells with My/Ery and My/Ly

potential in vitro. The intermediate population was predicted to contain equal proportions of cells with My/Ly and My/Ery output it was excluded from further analyses. As expected, with these new gates, Subset1 and 49f[+] Subset1 single cells produced significantly more colonies containing Ery cells than Subset2 and 49f[+] Subset2 single cells, whereas Subset2 and 49f[+] Subset2 single cells made significantly more colonies containing NK cells (Fig. 2b–c and Supplementary Fig. 3a, b). Specifically, both bilineage My/Ery and unilineage Ery colonies were produced in higher proportions by Subset1 and 49f[+] Subset1 single cells than by Subset2 and 49f[+] Subset2 single cells (Fig. 2d–e). Conversely, single cells from Subset2 and 49f[+] Subset2 generated more bilineage My/NK and unilineage NK colonies than Subset1 and 49f[+] Subset1 single cells (Fig. 2d–e). In all, 49f[+] Subset2 single cells also produced significantly less My colonies containing both monocytes and granulocytes colonies, but more containing exclusively monocytes than 49f[+] Subset1 (Supplementary Fig. 3c, d). As predicted, 49f[+] Subset1 cells divided on average, significantly later than 49f[+] Subset2 cells ($n = 1470$ single cells, Fig. 2f–g) indicating the novel-sorting strategies could also isolate populations with distinct kinetics of quiescence exit.

We then sought to confirm the lineage potential of cells within the Subset1 and Subset2 populations in a panel of other in vitro assays. In conditions that allow growth of B, NK and My cells[4], approximately 80% single cells from both populations produced a colony (Supplementary Fig. 3e). Subset2 single cells produced significantly more colonies containing B and NK cells than Subset1 single cells (Fig. 2h), indicating that the Ly differentiation potential of Subset2 is not restricted to the NK lineage. In semi-solid colony-forming unit (CFU) assay, Subset1 cells reproducibly generated more Ery-only and mixed (My/Ery) colonies than Subset2 cells (Fig. 2i). In line with this, in culture conditions mimicking human erythropoiesis[37] only Subset1 and 49 f[+] Subset1 cells could give rise to CD71[+]GlyA[+] Ery cells (Supplementary Fig. 3f). The same lineage differentiation properties of Subset1 and Subset2 single cells were also observed when these populations were sorted with distinct antibody clones and/or fluorochrome combinations (Supplementary Fig. 3g). In summary, these data demonstrate that single HSC/MPP and 49f[+] HSC cells with either My/Ly and My/Ery differentiation outputs in four different assays in vitro can be prospectively enriched based on distinct levels of cell surface expression of CD34 and CLEC9A.

**Initiation of lineage-priming programmes within 49f[+] HSCs.** To investigate the transcriptional status of 49f[+] Subset1 and 49f[+] Subset2 cells, we index-sorted 96 single cells of each population and performed single-cell RNA-seq (>78% of single cells passed quality control). Over the continuum of 49f[+] HSCs, 49f[+] Subset1

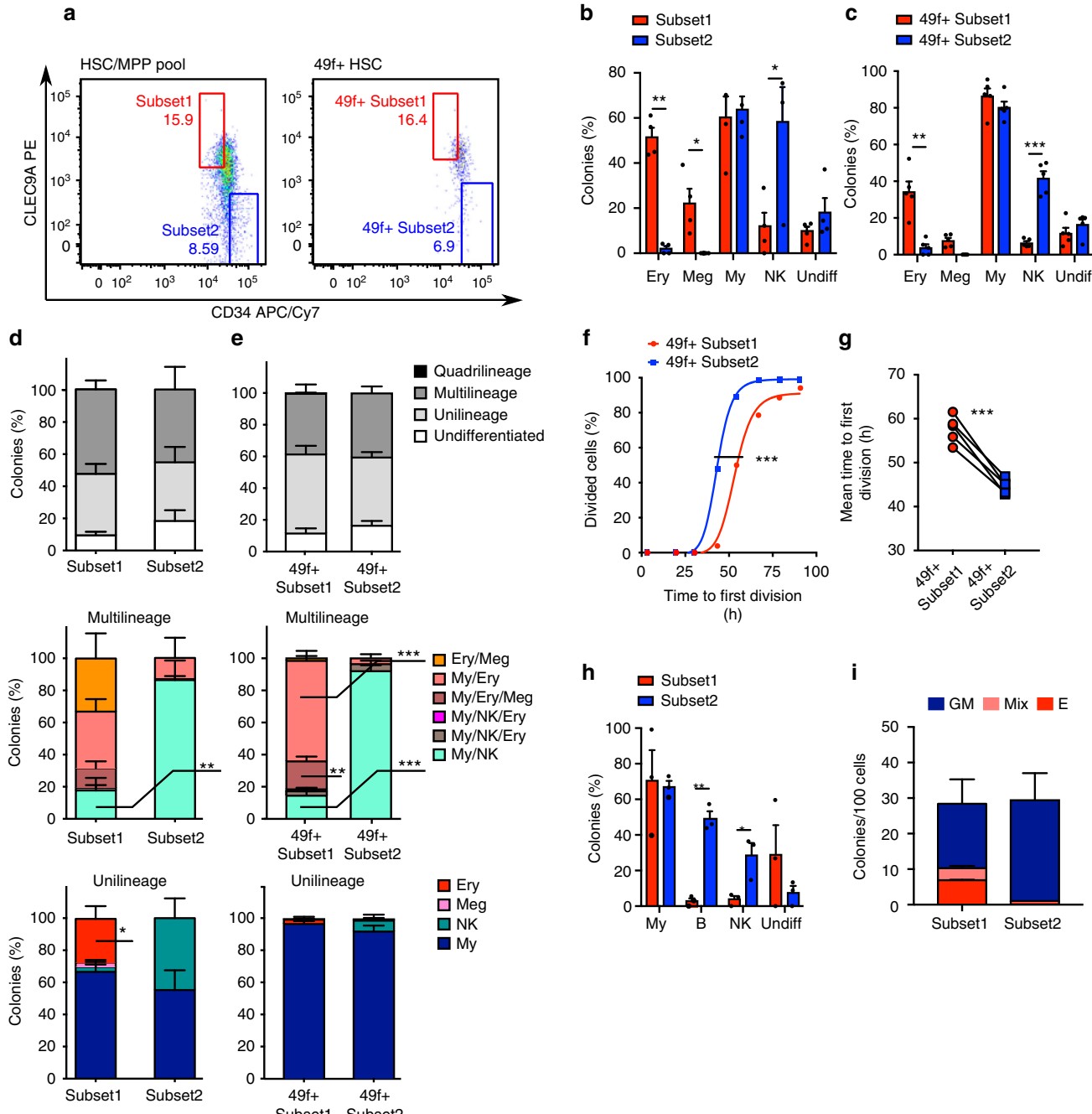

**Fig. 2** Prospective purification of HSC/MPP pool and 49f[+] HSC cells with distinct in vitro differentiation capacity. **a** Representative examples of the gating strategy derived from in silico analysis used to isolate the indicated populations. **b–e** Single cells were plated in the same assay as in Fig. 1a–f. Percentage of colonies containing differentiated cells of the indicated lineages, generated by single Subset1 (red) or Subset2 cells (blue) (**b**) and by single 49f[+] Subset1 (red) or 49f[+] Subset2 cells (blue) (**c**). Percentage of colonies of the indicated types derived from Subset1 and Subset2 (**d**) and from 49f[+] Subset1 and 49f[+] Subset2 single cells (**e**). $n = 4$ independent CBs, $n = 270$ colonies from Subset1, $n = 171$ colonies from Subset2 (**b**, **d**). $n = 5$ independent CBs, $n = 628$ colonies from 49f[+] Subset1, $n = 522$ from 49f[+] Subset2 (**c**, **e**). Mean ± SEM is shown *$p < 0.05$, **$p < 0.01$, ***$p < 0.001$ by two-sided paired $t$-test (**b–e**). **f** Representative example of cumulative first division kinetics of 49f[+] Subset1 (red) and 49f[+] Subset2 (blue) single cells. Curves are a non-linear least squares ordinary fit (non-log); ***$p < 0.001$ by extra sum of squares $F$-test. **g** Mean time to first division of 49f[+] Subset1 (red) and 49f[+] Subset2 (blue) (EC$_{50}$ of non-linear fit of cumulative first division kinetics as in **f**); $n = 5$ independent experiments each with an independent CB (with, respectively, 384, 384, 360, 150 and 192 single cells per experiment); ***$p < 0.001$ by two-sided paired $t$-test. **h** Cells were plated in an assay supporting differentiation of My, B and NK cells. Percentage of colonies generated by Subset1 (red) or Subset2 (blue) single cells and containing differentiated cells of the indicated lineages; mean ± SEM is shown, $n = 3$ independent CBs (Subset1 138 colonies, Subset2 142 colonies), *$p < 0.05$, **$p < 0.01$ by two-sided paired $t$-test. **i** Number of colonies/100 cells from either Subset1 or Subset2 populations plated in a CFU assay. The type of colony: erythroid (E), granulocyte and myeloid (GM) or a combination of both (mix) is shown, ($n = 2$ independent CBs); mean ± SEM is shown

and 49f+ Subset2 single cells formed two distinct clusters occupying opposite poles of the transcriptional space (Fig. 3a and Supplementary Fig. 4a–e). When integrated into the ICGS analysis of 49f+ HSCs (Supplementary Fig. 2g), three major clusters with roughly equivalent numbers of cells were found in the 49f+ HSC compartment (Fig. 3b and Supplementary Data 1). 49f+ Subset1 cells were significantly enriched in Cluster I (corresponding to Cluster I in Supplementary Fig. 2g), while 49f+ Subset2 cells were significantly enriched in Clusters IIa and IIb (collectively corresponding to Cluster II in Supplementary Figs. 2g and 3c). Cluster I was marked by the expression of genes previously associated to HSC function such as CXCR4, MAFF and JUND[38–40]. Cluster IIb, in contrast, was characterised by the expression of genes involved in mRNA metabolism (HNRNPC, PABPC1), glycolysis (LDHA), DNA replication (NAP1L1) as well as HLA molecules, indicating a more active state than Cluster I. Cluster IIa cells had mixed expression profiles from Cluster I and IIb (Fig. 3b). Consistently with data in Fig. 1k, 49f+ HSCs within Cluster I displayed significantly higher levels of CLEC9A and lower levels of CD34 at their cell surface than 49f+ HSCs present in Clusters IIa and IIb (Supplementary Fig. 4f). In summary, 49f+ Subset1 and 49f+ Subset2 cells are located in different areas of the transcriptional space of the 49f+ compartment, and differentially express a number of genes linked to HSC function and activation.

Given the differences in lineage differentiation in vitro of 49f+ Subset1 and 49f+ Subset2 cells, we hypothesised that lineage-priming programmes might already be established within the 49f+ HSC compartment. Gene Set Enrichment Analysis (GSEA) using population-specific gene signatures from highly purified human HSC and progenitor cells[31,32] and lineage-priming gene modules[18], showed that HSC-related genesets as well as genesets activated during Ery and Meg differentiation were enriched in 49f+ Subset1 cells compared to 49f+ Subset2 cells (Fig. 3d). Consistently, among genes with significantly higher expression in 49f+ Subset1 cells (FDR < 0.05; 96 genes, Fig. 3e and Supplementary Data 2), JUN, MECOM, MEIS1 and HIF1A have all been involved in HSC maintenance[39,41–43]. MLLT3, the earliest regulator of Ery-Meg differentiation reported to date[44], also showed higher expression in 49f+ Subset1. By contrast, Ly differentiation-related genesets were enriched in 49f+ Subset2 cells (Fig. 3d), which also expressed significantly higher levels of Ly genes such as HCST and JCHAIN (Fig. 3e). Mitochondrial (MT-CO2 and MT-RNR1) and cell cycle genes (CDK6, TFDP1, 35 genes total, Fig. 3e and Supplementary Data 2) were also expressed at significantly higher levels in 49f+ Subset2. As levels of CDK6 directly regulate the duration of HSC quiescence exit[31], the higher levels of CDK6 in 49f+ Subset2 cells provide a mechanism for the faster kinetics of quiescence exit observed (Fig. 2f–g). Strikingly, the number of significantly enriched lineage-priming genesets when comparing Subset1 and Subset2 cells, isolated from the HSC/MPP pool, was considerably larger than that observed within the 49f+ HSC compartment, indicating amplification of these programmes within the more heterogeneous HSC/MPP pool (Supplementary Fig. 4g and Fig. 3f–g, Supplementary Data 3). Hallmarks of cell cycle activation were also significantly enriched in Subset2 cells (Fig. 3h).

In conclusion, the 49f+ transcriptional landscape gradually progresses from a group of cells marked by stem cell and early Ery lineage-priming programmes (49f+ Subset1) to a group with established Ly-priming programmes and marks of HSC activation (49f+ Subset2). These programmes then become reinforced within the HSC/MPP pool.

**49f+ Subset1 cells hierarchically precede 49f+ Subset2 cells.** Given their strong HSC signature, we hypothesised that 49f+

Subset1 cells may give rise to 49f+ Subset2 cells. We thus sorted 49f+ Subset1 and 49f+ Subset2 cells in bulk (Fig. 4a, top panel) and after 5 days in culture, a total of 1152 index-sorted single cells derived from either 49f+ Subset1 or 49f+ Subset2 were placed in a secondary differentiation assay (as in Fig. 1a). At day 5, 49f+ Subset1 cells produced cells with a phenotype equivalent to that of day 0 49f+ Subset2, whereas the parental 49f+ Subset2 population could not produce any cells with a day 0 49f+ Subset1 phenotype (Fig. 4a, bottom panel).

In the secondary differentiation assay (Fig. 4b), all populations derived from 49f+ Subset1 cells (S1_1, S2_1 and Diff1) displayed higher clonogenic efficiency than populations derived from 49f+ Subset2 cells (S2_2 and Diff2, Supplementary Fig. 5a). Of note, Diff1 and Diff2 differed in the levels of CD45RA expression (Supplementary Fig. 5b). By day 5, 49f+ Subset1 cells maintained cells with similar differentiation properties as day 0 (S1_1, Figs. 2e and 4c, $p > 0.1$). These were also the only cells generating multilineage (Fig. 4c, top panel) and large colonies (Supplementary Fig. 5c, $p < 0.001$), suggestive of the most stem-like state. In all, 49f+ Subset1 cells also generated: (i) cells phenotypically and functionally undistinguishable from 49f+ Subset2 (S2_1, producing exclusively colonies with My and/or NK cells) and (ii) cells with predominant Ery but no Ly output (Diff1, Fig. 4c and Supplementary Fig. 5d). In contrast, amongst 49f+ Subset2 derived progeny, single cells from S2_2 produced colonies of small size (Supplementary Fig. 5c), qualitatively similar to those of Day 0 49f+ Subset2 cells, but a higher proportion of NK-cell containing colonies (Figs. 2e and 4c, $p < 0.001$, Supplementary Fig. 5d). Very few, mostly unilineage, colonies were obtained for Diff2 (clonogenic efficiency < 20%, Supplementary Figs. 5a and 4c). We conclude that 49f+ Subset1 cells are hierarchically placed above 49f+ Subset2 cells.

**Subset2 cells support durable My/Ly but not Ery engraftment.** The defining property of an HSC is its capacity to produce long-term multipotential engraftment in transplantation assays. Production of Ery cells in xenograft models is not reliable because of the lack of cross-reactivity between mouse EPO and the human EPO-receptor. To circumvent this limitation, we established a protocol in which xenotransplanted mice were injected intra-peritoneally with human EPO (hEPO) for more than 2 weeks before analysis. Mice treated with hEPO for at least 2 weeks consistently produced robust erythroid grafts when transplanted with saturating doses of HSCs, which was not the case in control mice (Fig. 5a). To ensure we could reliably detect small grafts in the Ery, My and Ly lineage, we used exacting flow cytometry standards (as in ref. [2] and see methods) and could reliably detect total engraftment levels above 0.01% (Supplementary Fig. 6a).

By flow cytometry, Subset1 and Subset2 contain similar proportions of phenotypic 49f+ HSCs (approximately 16–17%, Supplementary Fig. 6b). To measure the frequency of repopulating cells within Subset1 and Subset2, and to assess their lineage potential in vivo, we performed limiting dilution analysis (LDA) experiments at 2, 8 and 20 weeks post-transplantation, administering hEPO for up to 1 month before bone marrow analysis. Using the extreme limiting dilution analysis (ELDA) statistical method[45], we found no significant difference in the frequency of engrafting cells at 2 weeks or 8 weeks post-transplantation (Fig. 5b–c and Supplementary Table 1). However, at 20 weeks post-transplantation, we found that Subset1 contains 6-fold more long-term repopulating cells than Subset2, in which long-term repopulating cells were very rare ($p = 0.0004$, Fig. 5d and Supplementary Table 1).

At 2 weeks post-transplantation when engraftment is predominantly driven by progenitor cells, Subset1 cells produced

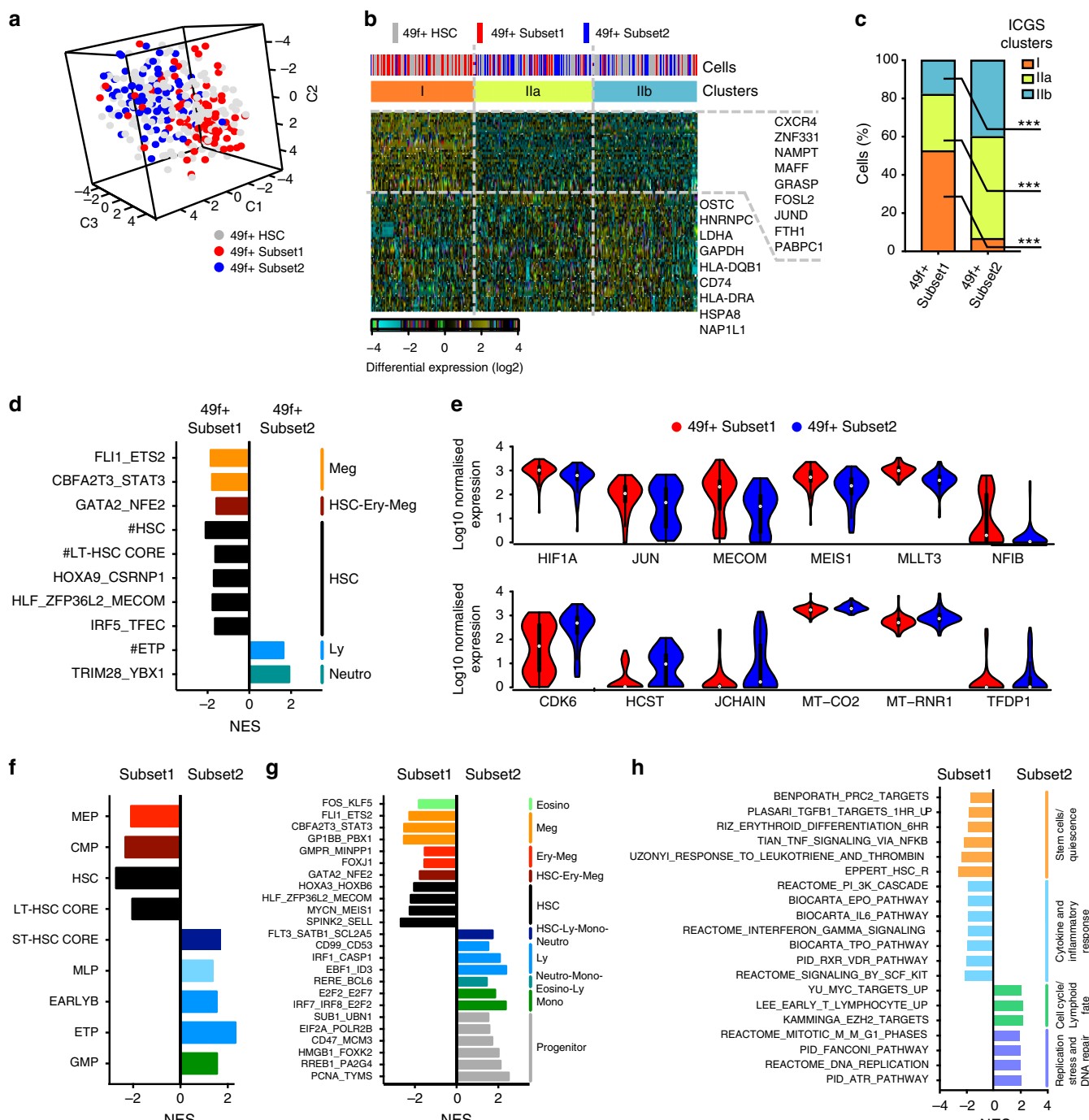

**Fig. 3** Polarisation of transcriptional profiles and establishment of lineage-priming programmes in the 49f+ HSC compartment. **a–e** Single-cell RNA-seq analysis of 49f+ HSCs (n = 169), 49f+ Subset1 (n = 78) and 49f+ Subset2 (n = 75) single cells. **a** tSNE representation of single 49f+ HSCs (grey), 49f+ Subset1 (red) and 49f+ Subset2 (blue) cells. All tSNE analyses were performed on highly variable genes (2420 genes computed as in ref. [58]). **b, c** ICGS analysis. **b** Heatmap of expression of guide genes selected by ICGS. Genes are represented in rows, with selected genes annotated on the side. Columns represent single cells, the phenotype of which is indicated by the uppermost bar of the top panel. Bottom bar of the top panel shows major clusters identified by ICGS. **c** Percentage of 49f+ Subset1 and 49f+ Subset2 single cells found in the indicated ICGS clusters; ***p < 0.001 by two-tailed Fisher test. **d** Population-specific signatures from[32], and lineage-priming gene modules derived from[18] significantly enriched (Benjamini-Hochberg adjusted p-value < 0.05 by pre-ranked GSEA) in 49f+ Subset1 and 49f+ Subset2 cells. **e** Log10 normalised expression of selected genes differentially expressed between 49f+ Subset1 (red) and 49f+ Subset2 (blue) cells (FDR < 0.05 by DESeq2). **f–h** 20-cell RNA-seq of Subset1 and Subset2. Significantly enriched (Benjamini-Hochberg adjusted p-value < 0.05 by pre-ranked GSEA) population-specific signatures from[32] (**f**), lineage-priming gene modules derived from[18] (**g**), and gene ontology terms and MSigDB genesets (**h**) in Subset1 and Subset2 cells. NES normalised enrichment score

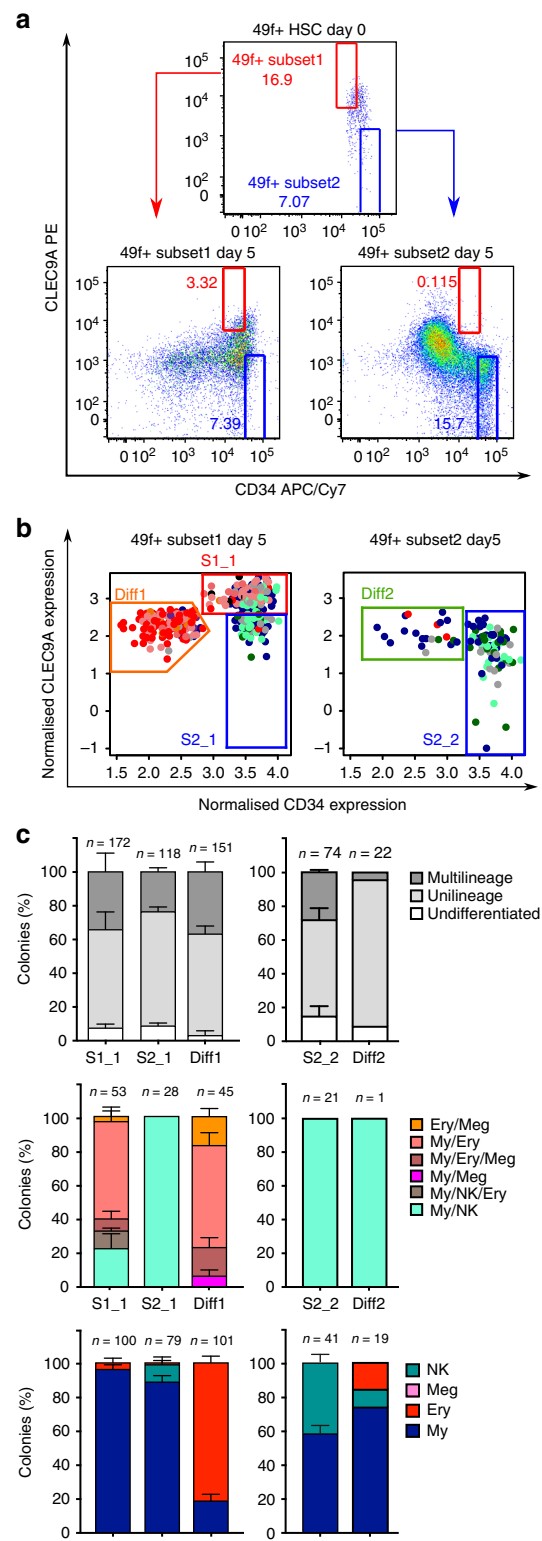

**Fig. 4** 49f+ Subset1 cells generate 49f+ Subset2 cells in vitro. **a** Sorting strategy used to isolate 49f+ Subset1 and 49f+ Subset2 populations at day 0 (top panel) and their derived populations after 5 days in culture (bottom panel). **b** Single cells from each of the indicated gates were sorted. Colours indicate the type of colony produced by each cell after 3 weeks of culture. $n = 576$ cells plated from three independent CBs for 49f+ Subset1 derived populations, $n = 576$ cells plated from three independent CBs for 49f+ Subset2 derived populations. **c** Percentage of colonies derived from single cells from the indicated day 5 populations. $n = 3$ independent CBs (except for Diff2 where $n = 1$). Mean ± SEM is shown. The total number of colonies analysed is shown for each population on the top of each bar

engraftment; Fig. 5g; or $p = 0.012$ when considering animals with human engraftment > 1%, Supplementary Fig. 6c and Supplementary Data 4). Only My and/or Ly cells were produced long-term in Subset2 transplanted animals (Fig. 5g). Altogether, these data show that long-term multilineage engraftment (Ly/My/Ery) is highly enriched in Subset1 cells. Importantly, Subset2 contains a small fraction of long-term repopulating cells, which exclusively differentiate towards the Ly and My lineages in this in vivo assay.

In addition, at 20 weeks post-transplantation, approximately 1:13 49f + Subset11 cells possessed long-term engraftment capacity, a frequency similar to unfractionated 49f+ HSCs[2]. In contrast, only 1:685 49f+ Subset2 cells ($p = 1.06 \times 10^{-10}$) could engraft up to 20 weeks after transplantation (Fig. 5h–i), a frequency similar to that observed for 49f− HSCs[2]. Consistently with data from Subset1 and Subset2, Ery cells were detected in the injected bone of 14 out of 19 mice engrafted with 49f+ Subset1 cells, but in none of the three mice engrafted with 49f+ Subset2 cells (Fig. 5j). Of note, in the NSG xenograft model the balance of mature lineages is highly skewed towards production of B cells. We nonetheless observed that three mice transplanted with 49f+ Subset1 displayed unusually low levels of Ly engraftment at 20 weeks post-transplantation (marked by arrows in Fig. 5j). This is suggestive of My-bias similar to that observed in mice[20].

Finally, we performed serial transplantation assays from primary animals injected with Subset1 or Subset2 cells. Seven out of 9 Subset1 and 2 out of 2 Subset2 secondary recipients were successfully engrafted (Supplementary Fig. 6d), indicating that both populations possess long-term repopulation capacity spanning a 32 week-period over serial transplantation. Collectively, Subset1, 49f+ Subset1, Subset2 and 49f+ Subset2 populations all contain relatively rare cells with long-term repopulating capacity, but with different lineage differentiation capacities (multipotent for Subset1 and 49f+ Subset1; erythroid-null myelo-lymphoid for Subset2 and 49f+ Subset2) and different frequencies.

**49f+ Subset2 and Subset2 cells are distinct from LMPPs.** 49f+ Subset2 and Subset2 repopulation capacity, albeit rare, extends well beyond that of My-Ly-restricted progenitors such as LMPPs and MLP, which do not engraft beyond 2 to 8 weeks after transplantation[4–6]. We thus hypothesised that the phenotypic 49f+ Subset2 and Subset2 compartments contain cells that are upstream of LMPPs and MLPs. To test this hypothesis, we compared the molecular profiles of these four populations (Supplementary Fig. 7a) by single-cell RNA-seq. Single cells were organised in a continuum, in which 49f+ Subset2 and Subset2 single cells overlapped, but LMPPs and MLPs were significantly shifted (Fig. 6a). LMPP cells were transcriptionally well distinct from both 49f+ Subset2 and Subset2 cells, differing from those by respectively 912 and 437 differentially expressed genes (Fig. 6b). In contrast LMPPs and MLPs were very similar (only 50 differentially expressed genes). Interestingly, a number of genes

small grafts that were either My or My/Ery, while Subset2 cells generated either Ly, My or My/Ly grafts (Fig. 5e), consistent with our results in vitro. At 8 weeks post-transplantation, Subset1 cells started producing Ly cells (Fig. 5f). Strikingly, at 20 weeks post-transplantation, 13 out of the 19 mice engrafted with Subset1 cells displayed multilineage (Ly/My/Ery) engraftment in the injected bone while none of the six mice transplanted with Subset2 cells did ($p = 0.0149$ by Fisher test considering mice at all levels of

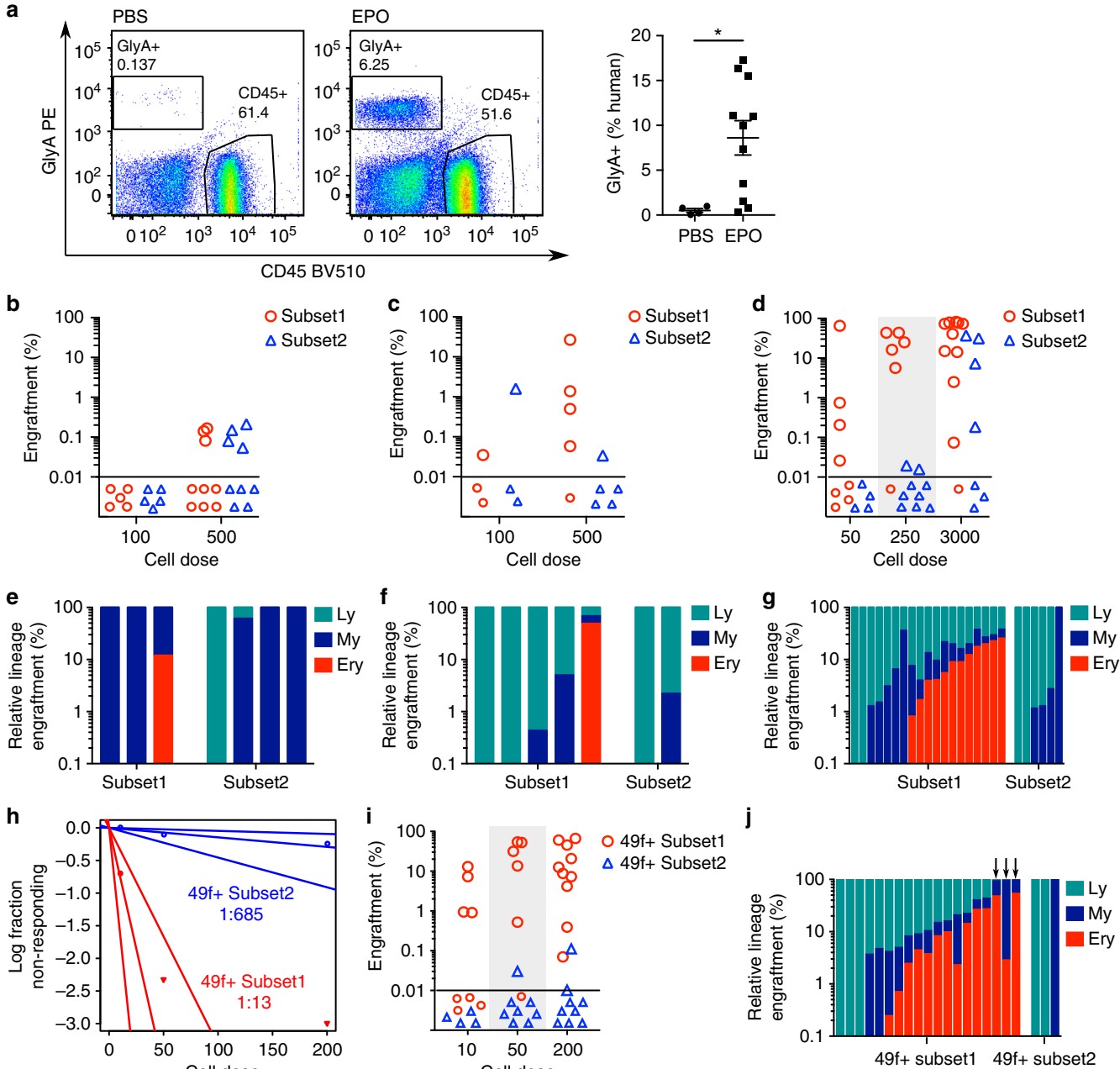

**Fig. 5** Distinct long-term repopulation and differentiation capacities of Subset1 and Subset2 cells in vivo. **a** Left: Representative flow cytometry plots from the injected femur of mice engrafted with CB CD34$^+$ cells and injected with seven doses of PBS (control) or EPO (20 units/injection). Right: relative Ery engraftment (percentage of total human engrafted cells), shown is mean ± SEM, *$p < 0.05$ by unpaired Mann–Whitney test. PBS: $n = 4$ mice, EPO: $n = 11$ mice. **b–d** Percentage of human engraftment (%CD45$^{++}$ + %GlyA$^+$) in the injected femur of mice transplanted with Subset1 (red) or Subset2 (blue) cells at 2 weeks (**b**, $n = 14$ transplanted mice per population), 8 weeks (**c**, $n = 8$ transplanted mice per population) and 20 weeks (**d**, $n = 25$ transplanted mice for Subset1 and $n = 20$ transplanted mice for Subset2) after transplantation. Dashed line: threshold of engraftment (%CD45$^{++}$ + %GlyA$^+$) ≥ 0.01 % and at least 30 cells recorded. Non-engrafted mice shown below dashed line. CD45$^{++}$ indicates cells positive for two distinct CD45 antibodies. **e–g** Distribution of differentiated cell types from each indicated lineage in the human graft of individual mice (injected femur, each bar represents one mouse) engrafted with Subset1 or Subset2 cells at 2 weeks (**e**), 8 weeks (**f**) and 20 weeks (**g**) after transplantation. My lineage: CD33$^+$ cells; Ly lineage: CD19$^{++}$ cells (positive for two distinct CD19 antibodies); Ery lineage: GlyA$^+$ cells. **e** Subset1: $n = 3$, Subset2: $n = 4$. **f** Subset1: $n = 5$, Subset2: $n = 2$. **g** Subset1: $n = 19$, Subset2: $n = 6$. **h** Estimation of frequency of long-term repopulating cells within 49f$^+$ Subset1 and 49f$^+$ Subset2 cells at 20 weeks post-transplantation by ELDA. **i** Percentage of human engraftment (%CD45$^{++}$ + %GlyA$^+$) in the injected femur of mice injected with 49f$^+$ Subset1 (red, $n = 24$ mice transplanted) or 49f$^+$ Subset2 (blue, $n = 21$ mice transplanted) cells at 20 weeks after transplantation. **j** Distribution of differentiated cell types belonging to each indicated lineage in the human graft of individual mice (injected femur, each bar represents one mouse) engrafted with 49f$^+$ Subset1 or 49f$^+$ Subset2 cells at 20 weeks after transplantation. 49f$^+$ Subset1: $n = 19$, 49f$^+$ Subset2: $n = 3$. Arrows indicate mice with unusually low levels of Ly engraftment

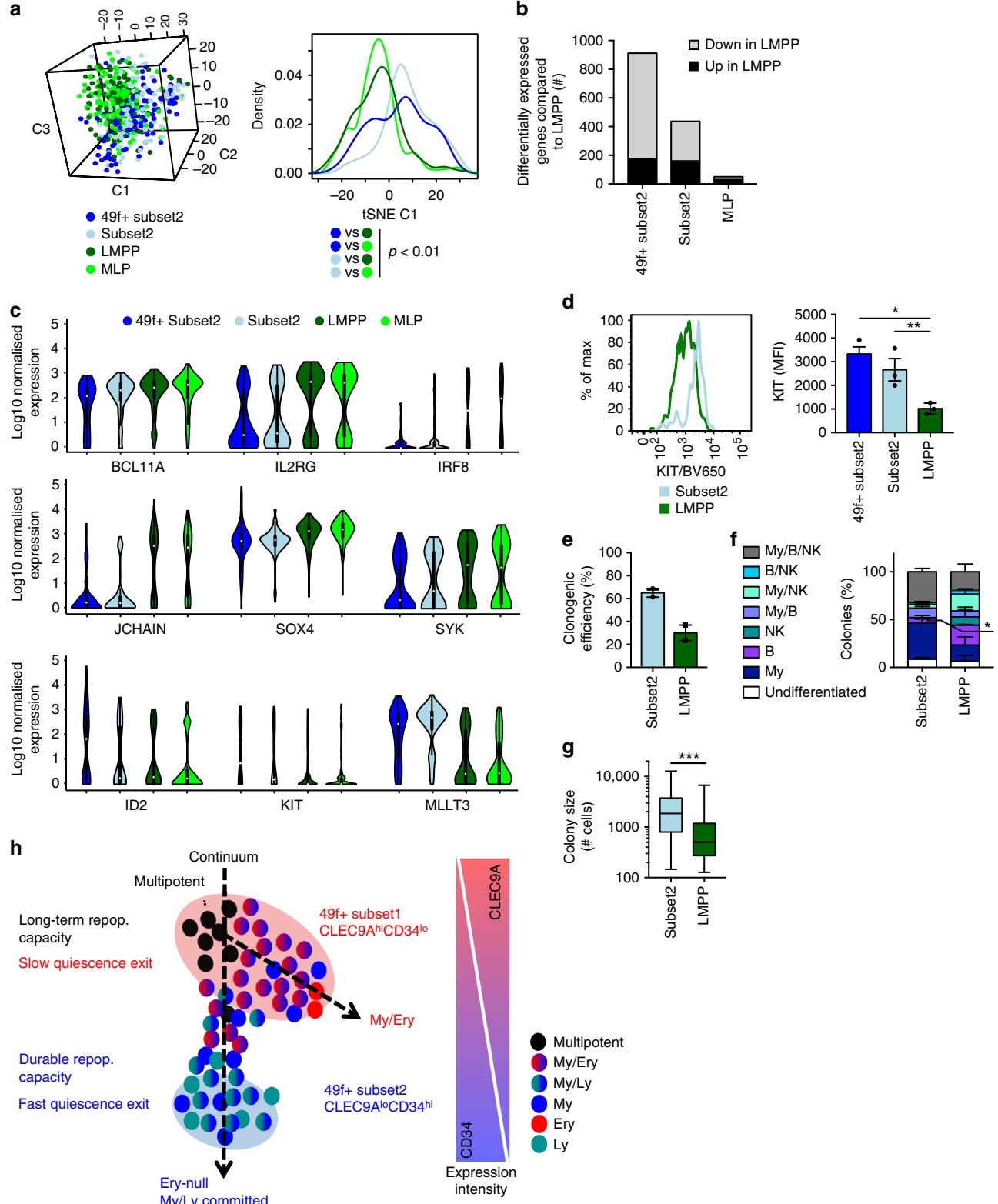

related to Ly-priming, including the *BCL11A* and *SOX4* transcription factors required for differentiation of MLPs to B cells[32], are gradually and significantly upregulated progressing from 49f+ Subset2 to Subset2 to LMPP to MLP (Fig. 6c). In parallel, genes involved in the maintenance of HSC self-renewal (*ID2*[46]) and engraftment capacity (*KIT*[34]) are significantly and progressively downregulated from 49f+ Subset2 to LMPPs/MLPs. Cell surface

levels of KIT protein, as well as those of other stem cell associated cell surface markers (Supplementary Fig. 7b–d), were also found to be significantly downregulated in LMPPs compared to 49f+ Subset2 and Subset2 cells (Fig. 6d). Finally, when the differentiation capacity of 240 Subset2 and 240 LMPP single cells along the My and Ly branches was tested in vitro, we observed that phenotypic Subset2 single cells had higher clonogenic efficiency

**Fig. 6** Molecular and functional differences between CD49f+ Subset2, Subset2 cells, LMPPs and MLPs. **a–c** Single-cell RNA-seq analysis of 49f+ Subset2 ($n = 119$), Subset2 ($n = 100$), LMPP ($n = 140$) and MLP ($n = 134$) single cells. **a** Left: 3D tSNE representation of indicated populations performed on highly variable genes (5831 genes). Right: density plot of the distribution of indicated populations along the first tSNE component. $p < 0.01$ by Kruskal–Wallis test with multiple comparison. **b** Number of significant differentially expressed genes between LMPP and the indicated populations (FDR < 0.05 by DESeq2). **c** $Log_{10}$ normalised expression of selected genes differentially expressed both between 49f+ Subset2 and LMPP and Subset2 and LMPP in single cells from indicated populations. **d** Cell surface expression of KIT on Subset2 and LMPP cells. Left: representative flow cytometry histogram, Right: quantification ($n = 3$ independent CBs, $*p < 0.05$ and $**p < 0.01$ by one-way ANOVA with Tukey's multiple comparison). **e–g** In vitro differentiation assay of Subset2 and LMPP single cells ($n = 240$ single cells from two independent CBs plated per population) in conditions supporting differentiation of My, B and NK cells. **e** Clonogenic efficiency of Subset2 and LMPP cells ($n = 2$ independent CBs). **f** Percentage of colonies of the indicated type derived from Subset 2 ($n = 153$ colonies from two independent CBs) and LMPP ($n = 72$ colonies from two independent CBs) single cells, $*p < 0.05$ by two-tailed paired $t$-test. **d–f** Shown is mean ± SEM. **g** Boxplots of the size of colonies generated by Subset2 ($n = 153$ colonies from two independent CBs) and LMPP ($n = 72$ colonies from two independent CBs) single cells, $**p < 0.01$ by two-tailed Mann–Whitney test. Boxplots show median, interquartile and range. **h** Graphical representation of the single-cell structure of the 49f+ HSC compartment proposed here

(Fig. 6e) and were overall less committed than phenotypic LMPPs, with significantly fewer single cells producing colonies containing only B cells, and no colonies containing only NK cells (Fig. 6f). Subset2 single cells were also more proliferative (Fig. 6g). We conclude that single cells within the 49f+ Subset2 and Subset2 compartments represent a developmental intermediate between a multipotent long-term repopulating stem cell and LMPPs (Fig. 6h). Ly lineage-priming (albeit at lower levels than in LMPPs) and loss of erythroid potential is already initiated in 49f+ HSCs giving rise to erythroid-null, myelo-lymphoid committed cells with infrequent durable reconstitution.

## Discussion

In this work, we analysed more than 6300 single cells from HSC/MPP and 49f+ HSC subsets with in vitro differentiation assays and single-cell RNA-seq. Our findings establish that the first lineage restriction events, which separate multipotent long-term repopulating cells from single cells with My/Ly but no Ery potential, occur within the phenotypic HSC/MPP and 49f+ HSC compartments, and not downstream of them, as classically thought. Several recent studies focusing on progenitors cells at single-cell resolution[3–6,11–13,15,17,18] indirectly suggested that lineage choices must occur to some degree either in the HSC or MPP compartment. Our study provides direct evidence that the developmental transition from true multipotentiality to bipotentiality occurs already within the phenotypic HSC/MPP compartment. We identify prospective purification strategies to highly enrich for the earliest fully erythroid-deficient myelo-lymphoid restricted cell type reported to date, and place it upstream of LMPPs. We thus propose to designate the latter lymphoid-primed short-term repopulating HSCs.

Our data considerably expands the current understanding of the cellular and molecular structure of the HSC/MPP pool in humans at single-cell resolution. A high degree of functional and molecular polarisation exists within the most immature 49f+ HSCs and the larger HSC/MPP pool compartments along an axis of anti-correlated cell surface expression of CLEC9A and CD34 markers. We observed a progression from the CLEC9A^hi CD34^lo extreme containing both rare cells with long-term repopulating multipotent capacity in vivo and cells with bilineage My/Ery potential in vitro, to the CLEC9A^lo CD34^hi extreme where single cells with restricted My/Ly potential in vitro and in vivo reside (Fig. 6h). Our data predict that cells with intermediate cell surface phenotypes (such as CLEC9A^lo CD34^lo and CLEC9A^hi CD34^hi) display heterogeneous behaviours that will need to be further explored. It is also important to note the limitations of in vitro differentiation assays that may not fully read-out all potentials. For example, many single cells throughout this continuum read-out exclusively myeloid in two distinct in vitro assays. It is possible that these cells would differentiate into dendritic cells in

other conditions, or could acquire different fates under stress. Interestingly, dendritic differentiation has recently been shown to branch out from the human HSC/MPP compartment[10]. Alternatively, myeloid differentiation may simply represent a default programme in the absence of activation of either Ly- or Ery-priming, as previously postulated[47].

One apparent paradox is that many distinct HSC behaviours have been identified but all cells with an HSC phenotype appear highly similar at the transcriptional level. Recent publications have put forward a model of a continuous and low primed HSC pool, from which progenitor cells with unilineage capacity originate[18]. Our data extend this model by establishing that distinct functional behaviours, in particular restriction to bilineage differentiation potential, are coordinated within the HSC/MPP pool by an underlying transcriptional structure. Despite the apparent continuity observed with most dimensionality reduction algorithms, and in contrast to previous studies[16,18], using the ICGS algorithm[36], we find that there are at least three distinct transcriptional states within the 49f+ HSC compartment. Two of these states are respectively enriched for the activity of specific transcription factors (such as MAFF, JUND, GATA2, HOXA9 and HLF in 49f+ Subset1) and for single cells with distinct read-out in in vitro and in vivo functional assays. We speculate that, given the resolution of single-cell RNA-seq, the HSC transcriptional landscape globally appears as a continuum because all phenotypic HSC subsets (described here and elsewhere[2,32,48]) share a common quiescent/metabolic signature, not seen in progenitor populations[29,32]. Our data demonstrate that at the extremes of this polarised continuum, identifiable and discrete changes in single-cell transcriptional programmes are associated with distinct behaviours of single phenotypic HSCs. This finding highlights the importance of developing robust algorithms to identify such discrete states and has important implications for the interpretation of large single-cell transcriptomics studies.

Unlike recent studies on human HSC populations[2,49,50], we administered human EPO to NSG mice to overcome the limitation in erythroid differentiation seen in this xenograft model[51]. We demonstrate that extending the range of lineage read-out to Ery differentiation significantly expands our understanding of heterogeneity within the phenotypic HSC/MPP pool, and identify rare multipotent (Ery/My/Ly) long-term repopulating HSC within the CLEC9A^hi CD34^lo fraction of the HSC/MPP and 49f+ HSC pools (Subset1). We hypothesise that these coexist with cells with restricted My/Ery potential (Fig. 6h; as suggested by experiments in vitro and in vivo at 2 weeks post-transplantation), which would share similarities with mouse MPP2[29,30] and Common Myeloid Repopulating Progenitors[28]. Nonetheless, as we cannot formally rule out that our assays underestimate the number of single cells with Ly potential in Subset1, future studies will have to address if truly My/Ery restricted or largely My/Ery

biased single-cell behaviours exist in this compartment. The small fraction of single cells from the HSC/MPP pool with Ery-only differentiation in vitro may correspond to the human CD71$^{hi}$-BAH-1$^{hi}$ erythroid restricted population described in[12]. In mouse, cells with highest megakaryocytic potential were found within the most immature phenotypic HSC subset[27,28,52]. Although our experimental strategy was not designed to identify putative human platelet-biased HSCs[27], given our transcriptional data and recent studies[12,18], it is possible that some single cells with this differentiation potential be located within Subset1 as well.

Importantly, without using the hEPO improved xenograft strategy, Subset2 could not be distinguished from multipotent 49f$^+$ HSCs. Like all Ly-restricted or Ly-biased HSC/MPP populations described so far in mice[22,24,29,30], Subset2 exhibit infrequent and limited repopulation capacity. This is in line with reports that transcription factor networks promoting Ly-priming also inhibit HSC long-term repopulation in human[32,46,53]. Even though the rare Subset2 cells with long-term repopulation capacity may not share the same molecular properties of the overall Subset2 population, we observed that acquisition of Ly-priming in Subset2 cells is accompanied by faster quiescence exit and other progenitor properties. The transition from multipotent cells to Ly-primed cells thus most likely involves changes in the molecular circuitry affecting self-renewal and repopulation capacity. Importantly, the Subset2 My/Ly-restricted cells identified here displayed repopulation capacity well beyond that reported for human LMPPs, MLPs or CD10-CD62L$^+$ cells[4–6,54]. In addition, they display lower Ly-priming at the functional and molecular level than LMPPs, indicating that they represent the very first step at which Ery potential is permanently lost and Ly potential specified (Fig. 6h). Finally, our data support a model of the haematopoietic hierarchy in which lineage commitment, particularly towards long-lived cells like lymphocytes, is specified very early in quiescent and durably engrafting cells, as opposed to shorter lived and/or higher cycling progenitors. This has important implications to understand the impact of ageing and pre-leukaemic mutations on blood formation in humans.

## Methods

**Human cord blood samples.** De-identified umbilical cord blood (CB) samples were obtained with informed consent from healthy donors through the Cambridge Blood and Stem Cell Biobank (CBSB) in accordance with regulated procedures approved by the Cambridgeshire Local Research Ethics Committee (07/MRE05/44 research study). CB samples were pooled independently of sex and processed as a single sample, with the exception of scRNA-seq experiments where single sex (males) CB samples were used.

**Human CB CD34$^+$ cells selection.** Mononuclear cells (MNCs) were isolated by Lymphoprep (Stem Cell Technologies) or Pancoll (PanBiotech) density gradient centrifugation of pre-diluted (1:1 with PBS) CB. The collected MNCs fractions were then depleted of red blood cells after 15 min incubation at 4 °C using Red Blood Cell Lysis Buffer (BioLegend) and CD34$^+$ cells were positively selected using the Micro Beads CD34$^+$ selection kit (Miltenyi Biotec) and AutoMACS cell separation technology (Miltenyi Biotec). CB CD34$^+$ cells were then frozen and stored at -150 °C until use.

**Sample preparation for flow cytometry sorting and analysis.** To separate different cell populations from CB CD34$^+$ cells, the cells were first thawed by dropwise addition of pre-warmed IMDM (Life Technologies) + 0.1 mg/ml DNase (Sigma) + 50% Fetal Bovine Serum (FBS, Life Technologies) and then incubated for 20 min in the dark at room temperature (RT) in PBS + 3% FBS antibody mix. Depending on the experiment, cells were stained with antibody panels A, B or I. For cell-sorting experiments, cells were sorted using BD FACS Aria Fusion or BD FACS Aria III sorters available at the NIHR Cambridge BRC Cell Phenotyping Hub facility. For single-cell-sorting experiments, cells were sorted with single-cell purity and index-sorting modes to allow retrospective correlation between the colony output from each single cell and its cell surface marker expression at the time of the sort. For bulk sorts, purity mode was selected. Purity for all sorts was > 95%.

The purity of sorts for Subset1, Subset2 and LMPP populations was precisely quantified and estimated to be > 99.95%. Specifically, when resorted, < 1 in 1700

Subset2 cells fell in the phenotypic LMPP gate, < 1 in 2900 Subset2 cells in the phenotypic Subset1 gate and < 1 in 8900 Subset1 in the phenotypic Subset2 gate.

Previously defined populations were sorted as follows: HSC/MPP pool (CD19$^-$ CD34$^+$ CD38$^-$CD45RA$^-$), 49f$^+$ HSCs (CD19$^-$CD34$^+$ CD38$^-$CD45RA$^-$CD49f$^+$ CD90$^+$), 49f$^-$ HSCs (CD19$^-$CD34$^+$ CD38$^-$CD45RA$^-$CD49f$^-$CD90$^-$), LMPP (CD19$^-$CD34$^+$ CD38$^-$CD45RA$^+$ CD10$^+$), MLP (CD19$^-$CD34$^+$ CD38$^-$CD45RA$^+$ CD10$^-$), GMP (CD19$^-$CD34$^+$ CD38$^+$ CD10$^-$CD7$^-$CD45RA$^+$) and CMPs-MEPs (CD19$^-$CD34$^+$ CD38$^+$ CD10$^-$CD7$^-$CD45RA$^-$), after gating out dead cells based on Zombie Aqua (Biolegend).

Flow cytometry analyses were performed using the BD LSR II Analyser with the BD LSR II HTC Analyser (BD Biosciences) in case of high-throughput analyses. Unstained cells and compensation beads (BD Biosciences) were used for compensation and as controls to set appropriate gates.

**Antibody panels**. Panel (A): CD45RA/FITC* (clone HI100, cat. no. 555488, 1 in 100), CLEC9A/PE (clone 8F9, cat. no. 353804, 1 in 75), CD49f/PECy5* (clone GoH3, cat. no. 551129, 1 in 100), CD38/PECy7 (clone HIT2, cat. no. 303516, 1 in 100), CD90/APC* (clone 5E10, cat. no. 559869, 1 in 100), CD19/Alexa700 (clone HIB19, cat. no. 302226, 1 in 300), CD34/APCCy7 (clone 581, cat. no. 343514, 1 in 100) and Zombie Aqua (cat. no. 423101, 1 in 2000); Panel (B): CD45RA/FITC*, CLEC9A/PE, CD49f/PECy5*, CD38/PECy7, CD90/APC*, CD19/Alexa700, CD34/ APCCy7, CD7/BV421* (clone M-T701, cat. no. 562635, 1 in 100), CD10/BV421* (clone HI10a, cat. no. 562902, 1 in 100), CD117 BV650 (clone 104D2, cat. no. 313222, 1 in 100) and Zombie Aqua; Panel (C): CD41/FITC (clone HIP8, cat. no. 303704, 1 in 1000), GlyA/PE* (clone HIR2, cat. no. 340947, 1 in 1000), CD45/ PECy5 (clone HI30, cat. no. 304010, 1 in 300), CD14/PECy7 (clone M5E2, cat. no. 301814, 1 in 1000), CD56/APC (clone HCD56, cat. no. 318310, 1 in 200), CD11b/ APCCy7 (clone ICRF44, cat. no. 301342, 1 in 300) and CD15/BV421 (clone MC-480, cat. no. 125614, 1 in 200); Panel (D): CD19/FITC (clone HIB19, cat. no. 302206, 1 in 200), CD45/PECy5, CD14/PECy7, CD56/APC, CD19/Alexa700, CD11b/APCCy7 and CD15/BV421; Panel (E): GlyA/PE*, CD33/PECy5* (clone WN53, cat. no. 551377, 1 in 1000), CD14/PECy7 and CD15/BV421; Panel (F): CD19/FITC, GlyA/PE*, CD45/PECy5, CD14/PECy7, CD33/APC* (clone P67.6, cat. no. 345800, 1 in 200), CD19/Alexa700, CD3/APCCy7 (clone HIT3a, cat. no. 300318, 1 in 100) and CD45/BV510 (clone HI30, cat. no. 304036, 1 in 500); Panel (G): CD19/FITC, CD71/PE (clone CY1G4, cat. no. 334106, 1 in 1000), CD45/ PECy5, GlyA/PECy7 (clone HI264, cat. no. 349112, 1 in 300), CD33/APC*, CD19/ Alexa700, CD14/APCCy7 and CD45/BV510; Panel (H): CD71/FITC (clone CY1G4, cat. no. 334104, 1 in 1000), GlyA/PE*, CD117/PECy5 (clone 104D2, cat. no. 313210, 1 in 1000), CD38/PECy7, CD36/APC (clone 5–271, cat. no. 336208, 1 in 300), CD45RA/Alexa700 (clone HI100, cat. no. 304120, 1 in 300), CD34/ APCCy7, CD105/BV421 (clone 43A3, cat. no. 323219, 1 in 300) and CD41a/ BV510* (clone HIP8, cat. no. 563250, 1 in 200); Panel (I): CD34/Alexa488 (clone 561, cat. no. 343619, 1 in 100), CD90/PE (clone 5E10, cat. no. 328110, 1 in 50), CD49f/PECy5*, CD38/PECy7, CLEC9A/APC (clone 8F9, cat. no. 353805, 1 in 100), CD19/Alexa700, CD45RA/BV521 (clone HI100, cat. no. 304130, 1 in 100) and Zombie Aqua.

All antibodies were from Biolegend except those indicated by * that are from BD. All antibodies were titrated and validated using appropriate positive and negative control samples from human blood mononuclear cells.

**My, Ly and Ery single-cell differentiation assay**. MS5 stromal cells were imported from Prof Katsuhiko Itoh at Kyoto University, and were verified to be Mycoplasma-free and were tested to support myeloid, erythroid, megakaryocyte and NK differentiation with appropriate control populations. MS5 cells were plated at passage 10–13 at 3000 cells/well in flat 96-well plates in 100 μl Myelocult H5100 medium (Stem Cell Technologies) + 1% Pen/Strep (Life technologies) one day before sorting HSC/MPP pool cells. On the day of the sort, the medium was changed to 100 μl/well MEM cytokine medium: StemPro medium with nutrients supplement (Life Technologies) supplemented with cytokines (SCF 100 ng/ml, Flt3-L 20 ng/ml, TPO 100 ng/ml, IL-6 50 ng/ml, IL-3 10 ng/ml, IL-11 50 ng/ml, GM-CSF 20 ng/ml, IL-2 10 ng/ml, IL-7 20 ng/ml; all Miltenyi Biotec), erythropoietin (EPO) 3 units/ml (Eprex, Janssen-Cilag), h-LDL 50 ng/ml (Stem Cell Technologies), 1% L-Glutamine (Life Technologies) and 1% Pen/Strep (Life Technologies). Indicated populations were index sorted as single cells (1 cell/well) and cultured for 2 weeks at 37 °C.

S1_1, S2_1, Diff1, S2_2, Diff2 were derived from bulk 49f$^+$ Subset1 and 49f$^+$ Subset2 cultured for 5 days in reduced MEM cytokine medium (lacking IL-11, IL-2 and IL-7). For culture of single 49f$^+$ HSCs, 49f$^+$ Subset 1, 49f$^+$ Subset2, S1_1, S2_1, Diff1, S2_2, Diff2 single cells were sorted into 96-well plates in 100 μl/well MEM cytokine medium and cultured for 3 weeks at 37 °C.

To monitor time of first division, cells were visualised and counted manually every 12 h for 4 days using an inverted microscope. The time of first division was recorded for each cell and all the empty wells were excluded from the experiment.

**My, B and NK cells single-cell differentiation assay**. One day before the sort, flat-bottom 96-well plates were coated with 100 μl/well 0.1 % gelatine (Sigma-Aldrich) for 2 h and then removed. MS5 stromal cells (below passage 7, tested to support myeloid, NK and B-cell differentiation from appropriate control populations) were plated at 4000 cells/well in 100 μl Myelocult H5100 supplemented with

SCF 100 ng/ml, Flt3-L 10 ng/ml, TPO 50 ng/ml, IL-6 20 ng/ml, GM-CSF 20 ng/ml, G-CSF 20 ng/ml, IL-2 10 ng/ml, IL-7 20 ng/ml, 1% Pen/Strep. On the day of the sort, single cells from the indicated populations were index sorted and cultured for 3 weeks at 37 ˚C.

**Single-cell differentiation assays analysis.** All single-cell-derived colonies were harvested into 96 U-bottom plates using a plate filter, in case of co-culture, to prevent the carryover of MS5 cells. Cells were then stained with 50 μl/well of antibody mix (antibody panel C for My, Ly and Ery differentiation assays or D for My, B and NK differentiation assays), incubated for 20 min in the dark at RT and then washed with 100 μl/well of PBS + 3% FBS. The type (lineage composition) and the size of the colonies formed were assessed by high-throughput flow cytometry. The colony output was determined using the gating strategy shown in Supplementary Fig. 1d. A single cell was defined as giving rise to a colony if the sum of cells detected in the $CD45^+$ and $GlyA^+$ gates was ≥ 30 cells. Ery colonies were identified as $CD45^-GlyA^+$ ≥ 30 cells, Meg colonies as $CD41^+$ ≥ 30 cells, My colonies as $[(CD45^+CD14^+) + (CD45^+CD15^+)]$ ≥ 30 cells, NK colonies as $CD45^+CD56^+$ ≥ 30 cells. Sub-classification of My colonies was done as following: Gran colonies were identified as $CD45^+CD15^+$ ≥ 30 cells and $CD45^+CD14^+$ ≤ 30, Mono colonies as $CD45^+CD15^+$ ≤ 30 cells and $CD45^+CD14^+$ ≥ 30 and Mono-Gran as $CD45^+CD15^+$ ≥ 30 cells and $CD45^+CD14^+$ ≥ 30 cells. In the B-cell assay, B cells were identified as $CD45^+CD19^{++}$ ≥ 30 cells, My colonies as $CD45^+CD11b^+$ ≥ 30 cells and NK colonies as $CD45^+CD56^+$ ≥ 30 cells. All high-throughput screening flow cytometry data was recorded in a blinded way, and at analysis correlation between the colony phenotype and originating population was only performed at the final stage.

**Index-sorting analysis.** All analyses below were performed with R (version 3.3.2). Information on cell surface marker levels from the indexed cells was recorded in.csv files during FACS sorting. When data was collected from different sorts occurring on different days, we used a normalisation pipeline established in ref. [16]. Briefly, data from each experiment were logicle transformed with the estimateLogicle function ($m$ = 5.1) of the FlowCore package, then normalised with the ComBat function from the sva package. Principal Component analysis was performed with prcomp function on the normalised data, and vector loadings were obtained from the prcomp rotation variable. Diffusion map dimensionality reduction was performed in the bglab package following methods described by Haghverdi et al.[55] using a local sigma informed by the 5 nearest neighbours.

**Colony-forming unit assay.** Indicated populations were sorted into 500 μl PBS + 3% FBS in 1.5 ml microcentrifuge tubes (400 cells/tube). Cells were then centrifuged at 500 × g for 5 min and re-suspended in 50 μl PBS + 3% FBS and added to 2.5 ml Methocult optimum (Stem Cell Technologies) with Flt3-L 10 ng/ml and IL-6 10 ng/ml (Miltenyi Biotec). Plating was performed according to the manufacturer's protocol in 6 well SmartDishes (Stem Cell Technologies). Plates were cultured 2 weeks at 37 °C and the number of each type of CFU (granulocyte-monocyte, erythroid or mixed) were determined by visual inspection from photographs taken with the StemVision instrument (Stem Cell Technologies) 14 days after plating. Colonies were harvested on day 14 and stained with the antibody panel E.

**Red blood cell assay.** Cultures mimicking human erythropoiesis were adapted from the conditions described by[37]. 350–1500 $49^+$ Subset1, $49^+$ Subset2, Subset1 or Subset2 cells were sorted and cultured for 7 days in Expansion medium (day 0–7), followed by a 9-day culture in Differentiation medium (Diff I: day 7–12, Diff II: day 12–16). Medium composition was as described before[37] and at analysis, cells were stained with the antibody panel H.

**Mice.** $NOD.Cg-Prkdc^{scid}Il2rg^{tm1Wjl}/SzJ$ (NSG) mice were obtained from Charles River or bred in-house. Experimental cohorts consisted of female age-matched animals (12–16 weeks of age at the time of transplantation). All animals were housed in a Specific-Pathogen-Free (SPF) animal facility and experiments were conducted under UK Home Office regulations. This research has been regulated under the Animals (Scientific Procedures) Act 1986 Amendment Regulations 2012 following ethical review by the University of Cambridge Animal Welfare and Ethical Review Body (AWERB).

**Xenograft transplantation assays.** NSG mice were sublethally irradiated (2.4 Gy). Twenty-four hours later they were anaesthetised with isoflurane and injected intrafemorally with the indicated doses of the specified populations, and subcutaneously with 0.1 mg/kg buprenorphine (Animalcare). No specific randomisation nor blinding was performed as all mice shared the same genetic background and were age-matched females. For transplantation at saturating doses of HSCs, 43,000 to 69,000 CB $CD34^+$ cells were used. For LDA experiments, all sorted subsets where maintained overnight in X-VIVO 10 medium (Lonza) supplemented with 1% BSA (Roche), 1% L-glutamine (Life Technologies), 1% Pen/Strep (Life Technologies), SCF 100 ng/ml, Flt3-L 100 ng/ml, TPO 50 ng/ml and IL-7 10 ng/ml before injection.

Mice transplanted with saturating doses of HSCs were given intraperitoneal injections of either PBS or hEPO (20 units/injection, Eprex, Janssen-Cilag) every other day for 2 weeks. For LDA experiments, mice analysed at 8 and 20 weeks post-transplantation received eight intraperitoneal hEPO injections (20 units/injection) in the 4 weeks prior to sacrifice.

At the indicated time point after transplantation, the injected femur was harvested and the bone marrow was flushed out. Bone marrow cells were stained using the antibody panels F or G.

Secondary transplantation was performed by purifying $CD34^+$ cells from individual primary animals using the Micro Beads $CD34^+$ selection kit (Miltenyi Biotec), then $CD34^+CD38^-$ cells were sorted by flow cytometry from each primary mouse. In all, 100–6000 cells were injected intrafemorally into secondary recipients. Animals were then sacrificed 12 weeks post secondary transplantation.

To ensure that we could confidently detect low levels of human engraftment, we used two distinct antibodies against CD45, and cells were considered human if positive for both ($CD45^{++}$). Mice were considered engrafted if (%$CD45^{++}$ + % $GlyA^+$) ≥ 0.01 % and at least 30 cells were recorded in these gates. We also stained the bone marrow of mice that were irradiated but not transplanted to define the background of our staining. None of these mice met the criteria for engraftment. For determination of the lineage composition of the human graft, we adopted the following parameters: My lineage: $CD45^{++}$ $CD33^+$ ≥ 20 cells; Ly lineage: $CD45^{++}$ $CD19^{++}$ ≥ 20 cells (positive for two distinct CD19 antibodies); Ery lineage: $CD45^-$ $GlyA^+$ ≥ 20 cells or $CD45^-$ $CD71^+$ $GlyA^+$ ≥ 20 cells.

**RNA sequencing.** The protocol used for the RNA-seq library preparation was adapted from the Smart-seq2 protocol of Picelli et al[56]. Single cells or 20 cells were sorted into 96-well PCR plates in a configuration determined to minimise putative batch effects due to position on the plate into 4 μl of lysis buffer prepared containing 0.4% (v/v) Triton X-100, 2 U/μl RNase inhibitor (Clontech), 5 mM DTT, 1 mM dNTP and stored at −80 °C. The ERCC (Ambion, Life Technologies) were diluted to a final concentration of 1:3,000,000. The PCR purification step was done with 20 μl of Ampure XP beads (ratio 1: 0.6/0.7, Beckman Coulter). The success of cDNA preparation was confirmed by optimal DNA signal detected by a 2100 Bioanalyzer with High-sensitivity DNA chip (Agilent). Illumina library preparation was carried out following the Illumina Nextera XT DNA sample preparation protocol. Library size distribution was checked on an Agilent high-sensitivity DNA chip and the concentration of the indexed library was determined using the KAPA library quantification kit (Kapa Biosystems). The sequencing was done using the Illumina Hiseq 4000 system.

**RNA-seq analysis.** Alignment was performed using GSNAP, and read counts were generated with HTseq. The same analysis pipeline has been used for both scRNA-seq and 20 cell samples. All analyses below were performed in R (version 3.3.2) with the bglab package ([https://github.com/wjawaid/bglab]). Briefly, quality control was performed blinded of the single-cell/sample group and single cells/samples were retained for further analysis if > $2 \times 10^5$ reads mapped to a gene feature, there were > 20% of genes over the total number of reads and < 20% of mitochondrial genes over mitochondrial + nuclear genes. Data were normalised for sequencing depth using size factor calculated on endogenous genes[57]. For the scRNA-seq, batch effects were corrected using the ComBat function from the sva package on all genes that had > 1 count. Highly variable genes (HVGs) were selected fitting a GAM model assuming a quadratic relationship between log coefficient of variance (CV) and log mean expression for ERCC spike-in genes, with the function techVar (bglab package) setting the MeanForFit parameter at $10^5$[58]. All dimensionality reduction techniques were applied to HVGs. Differential expression between the two populations was performed with DESeq2 using the doDESeq wrapper function of the bglab package[59]. Genes were considered differentially expressed if FDR < 0.05. Expression of selected genes was represented using violin plots (vioplot package) of the log10 (1 + normalised_counts).

ICGS analysis was run with AltAnalyze software v.2.1.0 ([http://www.altanalyze.org])[36] using normalised and batch corrected counts as input and default parameters. Cell cycle genes were excluded using the most stringent parameter. For the analysis including $49f^+$ Subset1 and $49f^+$ Subset2 single cells, the software identified six minor clusters that were regrouped into three major clusters.

For GSEA, genes were ranked by the DESeq2 statistic and pre-ranked GSEA was run using as genesets either the C2 subcollection of curated genesets, the C3 transcription factor target genesets from the MSigDB[60], population-specific signatures[31,32] or lineage-priming modules[18].

**Flow cytometry analysis.** Data were analysed with FlowJo software (v 9.9 or v10). Where large amount of samples were analysed, FlowJo data were exported and further analysed using R.

**Sample size determination and study design.** The maximum sample size given the availability of primary human samples and rarity of the populations studied was used in all experiments. For single-cell experiments, a minimum of 50 cells per population per individual biological replicate were determined to ensure sufficient power for the intended purposes. Unless otherwise specified, all single-cell differentiation experiments were repeated at least three times with independent

biological samples. For limiting dilution assay transplantation experiments, a minimum of eight animals per population over at least two cell doses analysed were used (range: 8–25; 2 or 3 cell doses). This was estimated from previous studies to be able to detect differences in long-term repopulating cell frequency of approximately fivefold[2].

**Statistical analysis.** Statistical analysis was performed with R (version 3.3.2) or Graph Pad Prism v7, after verification that statistical tests were appropriate given the distribution of the data and variance between groups.

**Code availability.** The bglab R code package used for all RNA-seq analysis is available at https://github.com/wjawaid/bglab. All other code is available from the authors upon request.

## Data availability

All single-cell RNA-seq data have been deposited in the GEO portal under the superseries accession number GSE115798, with two subseries: GSE104995 (49f[+] HSC, 49f[+] Subset1, 49f[+] Subset2, Subset1 and Subset2 datasets); GSE115639 (49f[+] Subset2, Subset2, LMPP and MLP datasets). All relevant data are also available from the authors.

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

## Acknowledgements

We thank the Cambridge NIHR BRC Cell Phenotyping Hub, particularly Anna Petrunkina-Harrison and Esther Perez for their flow cytometry advice; the Cambridge Blood and Stem Cell Biobank, specifically Joanna Baxter and the team of nurses consenting and collecting cord blood samples; David Kent for critical reading of the manuscript. E.L. is supported by a Sir Henry Dale fellowship from the Wellcome Trust (WT)/Royal Society. S.B. is supported by a CRUK Cambridge Cancer Centre PhD fellowship. Research in the E.L. and B.G. laboratories is supported by the WT, European Hematology Association, Cancer Research UK, Bloodwise, Medical Research Council (MRC), Biotechnology and Biological Sciences Research Council, NIH-NIDDK, and core support grants by the WT and MRC to the WT-MRC Cambridge Stem Cell Institute.

## Author contributions

S.B. and E.F.C. designed and performed experiments and edited the manuscript; V.C., A. S., N.M and W.W.Y.L. carried out experiments; B.P.S., E.D., K.Y.C.S., X.W. and W.J. performed bioinformatic analysis; B.G. supervised parts of the study and edited the manuscript; E.L. designed and supervised experiments and wrote the manuscript. S.B. and E.F.C. contributed equally to this work.

## Additional information

**Competing interests:** The authors declare no competing interests.

