## [Peer Review File · Nature Communications]

Reviewers' comments:

Reviewer #1 (Hematopoiesis, HSC)(Remarks to the Author):

In this manuscript the authors examine functional heterogeneity within the phenotypically defined human HSC compartment, and identify a subpopulation of LIN-CD34+CD38-CD45RA- cells, defined by low CLEC9A and high CD34 expression, that is restricted to myelo-lymphoid differentiation. This population is characterized molecularly using single cell RNA sequencing, and functionally in repopulation assays. The molecular and functional readouts are compatible with a multi-potent progenitor population with limited self-renewal.

For the most part the experiments are well performed (see comments below), and appropriately described. The key concern is that the observations made provide a fairly limited conceptual advance, since the functional characteristics of the CLEC9AloCD34hi sub-population is essentially those of the previously characterized lymphoid-primed multi-potent progenitor (Goardon et al 2011, cited), and it is quite likely that the authors have simply identified an overlap between the functional LMPP and phenotypic HSC definitions, an issue that is not addressed in the manuscript. The results therefore do not change our view of early lineage commitment in any significant way.

Specific comments:

1. The key concern, as mentioned above, is that the identified CLEC9AloCD34hi cells essentially represent an overlap with LMPPs (typically defined as LIN-CD34+CD38-CD45RA+CD10-). This could be addressed by comparing CD45RA, CLEC9A and CD34 expression in Subset2 and LMPP populations, and ideally also by molecular comparison (e.g. clustering/comparison of single cell transcriptomes). The current molecular analysis presented would support the similarity between LMPPs and Subset 2.

2. When discussing the transplantation data on Subset 2 it would be important to acknowledge that the rarity of repopulating cells within this fraction is so low that they could essentially be a contaminant. Numerous reports have shown that lymphoid bias in transplantation assays in the mouse is basically a result of HSC depletion. Are the Ly-biased outcomes from Subset 1 associated with lower overall reconstitution?

3. Also, when discussing these data it is fairly clear that the Subset 1 cells have no problems giving rise to lymphoid cells, and describing them as My-Ery restricted/enriched therefore seems inaccurate. For all intents and purposes they behave like multipotent HSCs, and their apparent in vitro lineage restriction could simply reflect that they give rise to lymphoid cells with slower kinetics than cells that are already lymphoid-primed (as supported by the fact that they take some time to generate cells with a Subset 2 phenotype in vitro). More optimal in vitro lymphoid conditions would be required to establish this.

4: While it is appreciated that some level of simplification can facilitate manuscript writing I think referring to Meg-E, myeloid and lymphoid cell types as three lineages is misleading, and in particular referring to clones with e.g. myeloid cells only as unilineage, is unwarranted, since there are multiple myeloid cell lineages. It should not be difficult to identify monocytes and neutrophils by FACS with the antibody panel used and describe the readouts accordingly.

Reviewer #2 (HSC, differentiation)(Remarks to the Author):

This is an elegant and well written paper describing a detailed analyses of human haematopoietic stem and progenitor cells. The study combines single cell index sorting, the analyses of differentiation capacity of sorted cells and single cell RNA sequencing to analyse the transcriptome of these cells. Through a series of elegant and well-presented experiments this paper identifies several novel findings that make this paper worthy of publication in Nature Communication.

It is certainly interesting to the haematopoiesis community but the approach and the analyses they have done would be relevant to other fields

Key findings:

- They identify a novel marker combination for HSPCs (defined by reciprocal levels of expression of the markers CD34 and CLEC9) using single cell sorting, functional analysis and single cell sequencing. They demonstrate that cells sorted based on the use of this new marker combination defines specific HSC 49f subpopulations with distinct functional outputs both in vitro and in vivo. Single cell sequencing defines the transcriptional profile of these cells. They define the hierarchical organization of two subsets with 49f+ Subset1 (CLEC9Ahi CD34lo) cells being able to produce cells with a phenotype equivalent to that of 49f+ Subset2 (CLEC9Alo CD34hi but 49f subset 2 cells were unable to give rise to 49f subset 1.
- The study describes a novel transplantation to identify progenitors with erythroid potential by injection of human EPO into transplanted immunodeficient mice. This allows identification of cell populations that were not able to be analysed in vivo previously. This in vivo model defines the functional output of their HSPC subsets, defined by CD34/CLEC9 expression.
- Most importantly the study provides convincing evidence that supports the proposal that lineage commitment occurs within the HSC pool rather than downstream of this this population as assumed previously. This adds to the growing body of evidence in the field that the haematopoietic hierarchy is more complex than the classical model.

The data they present supports their conclusions with appropriate statistical analysis performed on all the data.

Specific comments.

1. One caveat to their study is that the cells are cultured in vitro for a number of days prior to analyses and thus it is formally possible that their findings represents events within this culture model. They should formally state this as one of the limitations as it may not precisely recapitulate events in vivo. Nevertheless their findings will certainly inform attempts to generate blood cell lineages in vitro from sources such as iPSCs.

2. Lines 103-114 and Figure 1a,c. I am confused by the authors definition of unilineage and really don't understand why they have stated they include EryMeg (clearly bi-lineage) in that subset. Some clarification is required. The following statements seems contradictory-

"We therefore considered EryMeg colonies as unilineage colonies and focused on the three main lineages: My, Ly and Ery (including EryMeg). Only 3.5% of single HSC/MPP pool cells produced multilineage colonies comprising all three lineages (Fig. 1a). Approximately 50% of 110 the cells produced bilineage colonies, predominantly of the My/Ery type"

3. Figure 1e – clarification on how the data has been manipulated to generate this figure. Does each point represent the mean time for cells from a single cord? (n=4 but there are only 3 points).

4. The manuscript would benefit from having a schematic 'haematopoietic hierarchy' figure that summarises their findings.

Reviewer #3 (System biology, RNA-seq)(Remarks to the Author):

The authors analysed single cells from human haematopoietic stem cell (HSC) subsets with differentiation assays and single-cell RNA-seq. They observed that CLEC9A and CD34 cell surface expression correlates with lineage potential and HSC transcriptome profiles. They provide evidence that first lineage restriction events occur already within the HSC compartment to generate myeloid-committed cells with no erythroid differentiation capacity.

Up to 8 cell surface markers have been used to distinguish HSC subsets with predetermined functional polarisation. It is an interesting study, but as I'm not a researcher in this particular field, I can not fully appraise the relevance. The textbook model of haematopoiesis involves a series of discrete cell stages. At a closer (single-cell) look, this is rather a continuous process. In this regard, this study complements novel findings published by others, and it is also likely that future studies will identify additional markers to further delineate in even more detail when and how distinct lineages emerge.

Belluschi et al. used appropriate technologies and tools for data analysis. The manuscript is well written and includes recent literature. The conclusions seem to be supported by the data. I have few comments that may help to improve the paper.

1. In several assays, the HSC/MPP pool and the 49f+ HSC subset showed similar outcomes. I wonder whether this was previously anticipated by the authors. They may better emphasize the differences (to readers not familiar with previous papers), e.g., as those seen in Suppl Table 3.

2. Abstract [page 1, line 17]: add CD19-.

3. Results: a) [page 5]: add percentages of 49f+ and 49f- cells in HSCs (only mentioned in the Suppl).

b) [page 5, line 94] why was CD117 included?

c) 49f+: the second marker CD90 could be mentioned more often.

4. Fig 1d: I assume that the boxes are wrongly labelled (My/Ery <=> My/NK) – please check. Moreover, there are two identical significance lines below the boxplot for PC2. Please check whether the numbers given in the legend [page 40, line 914 and line 917] and in the Results [page 5, line 91] are correct: "435 cells/colonies from 6/5 CB samples."

5. Fig 5f / 5j: n=2 for Subset2 is a bit critical. If possible, increase sample size to provide more confidence in the results. Otherwise, justify why only 2 mice were used.

6. Suppl Fig 2a (right one): I assume that the boxes are wrongly labelled (My/Ery <=> My/NK).

7. Suppl Table 2: CD34 and CLEC9A are not in this table, which is not intuitive – please discuss and report fold-changes for these genes.

8. Suppl Table 3: What was the statistical test to compare repopulating capacity?

9. Discussion: a) [page 18, line 406-8]: Is it really justified to write "whole" and "unipotentiality"?

b) [page 20, line 463]: define abbreviations MPP2 and CMRPs.

c) The authors focused on extreme cell populations. However, they may also speculate (more directly) about CD49f+CD90- and CD49f-CD90+ cells and CLEC9A^{lo}CD34^{lo} and CLEC9A^{hi}CD34^{hi} cells, which

represent fairly large fractions.

d) The authors should discuss transcription factors in more detail (e.g., those identified by GSEA analysis and those reported in Suppl Tables 1 and 2).

10. Methods: Please proofread and revise some minor typos. Check plural forms, consistency in terminology, definition of relevant abbreviations, formatting.

11. The single-cell RNA-seq data have been deposited in GEO. However, the data are still private (no token provided). I could not check whether the experimental methods are described in sufficient detail in these GEO records. The authors should add the 20-cell RNA-seq data as GEO SubSeries.

Altogether, we believe these new changes substantially improve our manuscript and thank all Reviewers for their constructive comments. Importantly, all the conclusions we had made in our original manuscript have been validated by the new experimental data and analysis. We are grateful to the Reviewers for all their suggestions. **In addition, the direct comparison to LMPPs now provides direct evidence that our approach has identified a cell behaviour that has not been previously reported (neither in the HSC nor progenitor compartment): rare cells from the human phenotypic CD49f⁺ compartment, which can provide durable and serially transplantable grafts, restricted to the myeloid and lymphoid lineage, in the absence of erythroid cell production. Our results place these cells upstream of LMPPs, thus identifying the earliest step at which erythroid commitment is lost and lymphoid priming established.** We believe this result will be of significance to the community.

In blue below are the Reviewers' comments. Our detailed response to each one of them is given in black, and we have highlighted in bold the changes we have made to the manuscript to address the Reviewers' concerns.

Reviewers' comments:

Reviewer #1 (Hematopoiesis, HSC)(Remarks to the Author):

In this manuscript the authors examine functional heterogeneity within the phenotypically defined human HSC compartment, and identify a subpopulation of LIN⁻ CD34⁺ CD38⁻ CD45RA⁻ cells, defined by low CLEC9A and high CD34 expression, that is restricted to myelo-lymphoid differentiation. This population is characterized molecularly using single cell RNA sequencing, and functionally in repopulation assays. The molecular and functional readouts are compatible with a multi-potent progenitor population with limited self-renewal. For the most part the experiments are well performed (see comments below), and appropriately described. The key concern is that the observations made provide a fairly limited conceptual advance, since the functional characteristics of the CLEC9A^{lo}CD34^{hi} sub-population is essentially those of the previously characterized lymphoid-primed multi-potent progenitor (Goardon et al 2011, cited), and it is quite likely that the authors have simply identified an overlap between the functional LMPP and phenotypic HSC definitions, an issue that is not addressed in the manuscript. The results therefore do not change our view of early lineage commitment in any significant way.

Specific comments:

1. The key concern, as mentioned above, is that the identified CLEC9A^{lo}CD34^{hi} cells essentially represent an overlap with LMPPs (typically defined as LIN⁻ CD34⁺ CD38⁻ CD45RA⁺ CD10⁻). This could be addressed by comparing CD45RA, CLEC9A and CD34 expression in Subset2 and LMPP populations, and ideally also by molecular comparison (e.g. clustering/comparison of single cell transcriptomes). The current molecular analysis presented would support the similarity between LMPPs and Subset 2.

We thank the Reviewer for qualifying our experiments as well performed. We understand the Reviewer's concern that our 49f⁺ Subset2 and Subset2 (CLEC9A^{lo}CD34^{hi}) populations may be equivalent to lymphoid-primed multi-potent progenitor (LMPPs), as cells within these 3 phenotypic populations are incapable of producing erythroid cells but can make cells of both the myeloid and lymphoid lineage. However, we found a very clear difference in repopulating capacity of the CLEC9A^{lo}CD34^{hi} populations and LMPPs upon transplantation. In all publications that to our knowledge describe LMPPs (Goardon et al., Cancer Cell, 2011; Karamitros et al., Nature Immunology, 2018), there is no evidence that cells from that population can engraft into NSG mice for more than 8 weeks even

at high doses (up to 10,000 cells). In contrast as few as 250 Subset2 and 50 49f⁺ Subset2 cells could produce a graft for up to 20 weeks as we had originally shown in our Figures 5d and 5i. We had commented on these important differences both in our results and discussion section in our original manuscript.

We do nevertheless concur that a direct head to head comparison of the molecular and functional properties of Subset2 populations and LMPPs would provide important additional information to strengthen our manuscript. We have thus performed the following experiments, which overall show clear molecular and functional differences between previously reported LMPPs and our newly identified 49f⁺ Subset2 and Subset2 cells:

- 1) According to the reviewer's suggestion, we have compared the cell surface expression of CD45RA, CLEC9A, CD34 as well as KIT on LMPPs, 49f⁺ Subset2 and Subset2 cells. By definition, LMPPs are CD45RA⁺ while 49f⁺ Subset2 and Subset2 are CD45RA⁻, but we nonetheless present the data in view of the Reviewer's request. For CD34 we did not observe any significant differences between these 3 populations. For CLEC9A we however found that LMPPs expressed significantly lower levels than 49f⁺ Subset2 cells. Finally, we also found that LMPPs had significantly lower cell surface expression of KIT, a marker that in primitive human haematopoietic cells directly correlates with more efficient and robust engraftment (Cosgun et al., Cell Stem Cell, 2014). This is consistent with our findings that cells within the 49f⁺ Subset2 and Subset2 populations can engraft NSG mice long-term and suggests that 49f⁺ Subset2 and Subset2 are less differentiated than LMPPs.

Data for KIT cell surface expression in 49f⁺ Subset2, Subset2 and LMPPs is now presented in Figure 6d, while data for CD45RA, CD34 and CLEC9A can be found in Supplementary Figure 7b-d.

- 2) We generated an entirely new batch of single-cell RNA-seq profiles for 49f⁺ Subset2, Subset2, LMPPs and MLPs (Doulatov et al., Nat. Immun., 2010; CD34⁺CD38⁻CD45RA⁺CD10⁺). A total of 493 cells passed quality control (49f⁺ Subset2: 119 cells; Subset2: 100 cells; LMPP: 140 cells; MLP: 134 cells). We observed that these 4 populations are placed in a single cell continuum. This is not surprising given all single-cell RNA-seq approaches from other groups have also placed known populations across the human haematopoietic hierarchy in a continuum (Velten et al., NCB, 2017; Karamitros et al., Nat. Immun., 2018). Importantly, LMPPs are well distinct from 49f⁺ Subset2 and Subset2 (912 and 437 genes are differentially expressed between LMPPs and 49f⁺ Subset2 and Subset2 respectively), but much less so from MLPs (only 50 genes were found to be differentially expressed between LMPPs and MLPs). We also observe that a number of genes related to Lymphoid priming are gradually and significantly upregulated progressing from 49f⁺ Subset2 to Subset2 to LMPP to MLP. These include genes such as IL2RG, JCHAIN and SYK but also the BCL11A and SOX4 transcription factors, that we have shown in the past to be upstream of the self-regulatory EBF1/PAX5 loop and required for differentiation of MLPs to B cells (Laurenti et al., Nat.Immun., 2013). Similarly, the IRF8 transcription factor, associated with myeloid and dendritic cell differentiation, is also significantly upregulated in LMPPs and MLPs compared to 49f⁺ Subset2 and Subset2 cells. Finally, we also observe that genes involved in the maintenance of HSC self-renewal (ID2; Van Galen, Cell Stem Cell, 2014) and engraftment capacity (KIT; Cosgun et al., Cell Stem Cell, 2014) are significantly downregulated in LMPPs and MLPs compared to 49f⁺ Subset2 and Subset2 cells.

Altogether, these new molecular data show that LMPPs are clearly distinct from 49f⁺ Subset2 and Subset2 cells at the transcriptional level and are consistent with a model whereby LMPPs are more differentiated cells than 49f⁺ Subset2 and Subset2 cells. **The analysis of these novel scRNA-seq experiments is now presented in Figure 6a-c, and all the new single cell transcriptome data has been deposited to GEO, to facilitate access by the wider research community. Reviewers can access it at this accession number (GSE115639) by using the following token: ivotysoivsdpwv.**

- 3) To further validate the notion that LMPPs are downstream of Subset2 cells, we have also performed functional single cell differentiation assays in conditions conducive to simultaneous differentiation of Myeloid, B and NK cells (same assay that we used to compare the lymphoid output of Subset1 and Subset2 presented in Figure 2h). Here we purified a total of 240 single LMPPs and 240 Subset2 cells from 2 independent cord blood samples and subjected them to this assay. Our results indicated the clonogenic efficiency of Subset2 was substantially higher than that of LMPPs and that importantly the type of colonies produced was significantly different. Specifically, significantly more single LMPPs produced colonies containing only B cells, and only single LMPPs produced colonies containing only NK cells. The size of the colonies derived from single LMPPs was also significantly smaller than those derived from single Subset2 cells. Collectively, these data show that the phenotypic LMPP compartment contains significantly more single cells that are fully restricted to the lymphoid lineage and are less proliferative than the phenotypic Subset2 compartment, indicating that as a population LMPPs are more differentiated than Subset2 cells. **This new functional data directly comparing single cells from the LMPP and Subset2 compartment is now presented in Figure 6e-g.**

Finally, to further verify that the engraftment we observed at 20 weeks post-transplantation from Subset2 cells was not just due to lingering long-lived lymphoid and myeloid cells in the absence of self-renewing stem cells, we have performed secondary transplants. 7 out of 9 Subset1 and 2 out of 2 Subset2 secondary recipients transplanted with primary mice sacrificed at 20 weeks post-transplantation were successfully engrafted for another 12 weeks. These new data demonstrate that Subset2 cells can sustain human engraftment in the NSG model over serial transplantation and a total of 32 weeks, **and this data has been added to the results section of our revised manuscript (Suppl. Figure 6d)**. As discussed above, this is incompatible with the engraftment potential reported in the literature from “contaminating LMPPs”.

In conclusion, we thank the Reviewer for their constructive criticism as collectively, these new experiments demonstrate that at single cell level, the Subset2 phenotypic compartment contains more single cells with bipotential myelo-lymphoid capacity, and with a higher proliferative capacity than the LMPP compartment. In addition, Subset2 single cells are transcriptionally distinct from LMPPs, marked by higher level of stem cell genes, and while lymphoid-priming at the transcriptional level has initiated in these cells, it is at lower levels than in the vast majority of phenotypically defined LMPPs. **Based on these experiments we now conclude that single cells within the Subset2 compartment represent a developmental intermediate between a multipotent long-term repopulating stem cell and LMPPs.**

Our data thus does not support a model in which rare LMPPs are found within the phenotypic HSC compartment. Rather we demonstrate that within the human 49f⁺ and HSC/MPP pool compartments, there are single cells, that can be highly enriched by using the CD34^{hi}CLEC9A^{lo} purification strategy, in which Erythroid potential has been lost, and the molecular programme of restriction to the Lymphoid and Myeloid lineages has been initiated (albeit to a lower level than in LMPPs). **Importantly, in the 49f⁺ Subset2 and Subset2 compartments, there are infrequent cells in which self-renewal and repopulation capacity are maintained to a higher degree than in any non-fully multipotential haematopoietic cell previously reported in humans. These findings are now reported in the results (Figure 6), discussion and abstract of the revised manuscript, and represent a major message of our manuscript.**

2. When discussing the transplantation data on Subset 2 it would be important to acknowledge that the rarity of repopulating cells within this fraction is so low that they could essentially be a contaminant.

For transparency in the manuscript, we have added a sentence clarifying that long-term repopulating cells are rare in Subset2 (line 361).

To investigate whether the engraftment observed could potentially originate from a contaminant, we precisely quantified the purity of our sorts for Subset1, Subset2 and LMPPs. Across all our sorts, we estimated that a maximum of 1 in 1700 Subset2 cells would fall in the LMPP phenotypic definition. At our intermediate LDA dose (250-500 cells) it is thus extremely unlikely that any LMPPs were transplanted. At the highest dose we injected (3000 cells) we thus expect 1 or maximum 2 LMPPs to be transplanted. Given previous reports have failed to see engraftment from up to 10000 LMPPs up to 8 weeks post-transplantation (Goardon et al., Cancer Cell, 2011), we find it extremely unlikely that the engraftment levels ranging from 0.19% to 34.4% observed at 20 weeks post-transplantation in 4 distinct mice transplanted with high doses of Subset2 cells would originate from 2 contaminant LMPPs.

Another possibility is that these levels of engraftment derive from contaminating Subset1 cells. We estimated that across all of our sorts, less than 1 in 2900 Subset2 cells would fall in the Subset 1 phenotypic definition. So even at the highest dose used in this study, it is extremely unlikely that even one Subset1 cell may have been transferred and would give rise to myeloid or lymphoid engraftment, without Ery differentiation.

The methods section now includes these new estimations of flow cytometry purity that exclude potential contamination of the Subset2 compartment with cells from the phenotypic Subset1 or LMPP compartments.

Numerous reports have shown that lymphoid bias in transplantation assays in the mouse is basically a result of HSC depletion. Are the Ly-biased outcomes from Subset 1 associated with lower overall reconstitution?

The reviewer is right that in mouse models of HSC transplantation, once HSCs are depleted, there are lingering lymphoid cells that can be interpreted as Ly-biased outcomes. In the NSG xenograft model however the situation is somewhat different as the environment is strongly supporting B cell development. As a result, the “homeostatic situation” is what would be considered Ly-bias in mouse models.

Nonetheless, we agree that it is important to verify how lineage output varies as a function of overall engraftment. For this, we have thus plotted the relative Ly, My and Ery engraftment as a function of the overall human engraftment for each mouse in this study. As can be seen in the graphs below, when engraftment is low, it is more likely that mice are engrafted with only one lineage, but this is not necessarily Ly, as in approximately 25% of mice engrafted at levels lower than 1%, engraftment is purely My. This occurs similarly with mice transplanted

with cells from Subset1 and Subset2 and is consistent with what was observed in previous studies using the NSG model (albeit without hEPO treatment; Notta et al., Science, 2011).

Figure 1: Relative engraftment (percentage of total human grafted cells) of the Ly (left), My (middle) and Ery (right) plotted as a function of human engraftment (percentage of CD45+ and GlyA+ present in the injected bone of each transplanted mouse). Subset1: n=27 mice; Subset2: n=12 mice; 49f+ Subset1: n=19 mice; 49f+ Subset2: n=3 mice.

Importantly, we have also verified that our findings on the different lineage output of Subset1 and Subset2 hold true when restricting our analyses exclusively to mice with engraftment >1%. Indeed, for Subset1 transplanted animals, 13/15 animals engrafted at >1% produced Ery cells, while 0/3 animals transplanted with Subset2 did (p=0.012 by Fisher test).

So even when restricting our analyses to animals with human engraftment >1%, our original conclusion that Subset2 cannot differentiate along the Ery lineage, whereas Subset1 can, is confirmed. We have now added a sentence in the results section to highlight this (line 373), and the plots above have been included in Suppl. Fig. 6c.

3. Also, when discussing these data it is fairly clear that the Subset 1 cells have no problems giving rise to lymphoid cells, and describing them as My-Ery restricted/enriched therefore seems inaccurate. For all intents and purposes they behave like multipotent HSCs, and their apparent *in vitro* lineage restriction could simply reflect that they give rise to lymphoid cells with slower kinetics than cells that are already lymphoid-primed (as supported by the fact that they take some time to generate cells with a Subset 2 phenotype *in vitro*). More optimal *in vitro* lymphoid conditions would be required to establish this.

We agree with the Reviewer that stating that all cells in Subset1 are “My/Ery restricted” would be inaccurate. But from careful examination of our original manuscript, we never intended to use the term “My/Ery restricted” to label the whole Subset1 population but rather to qualify the colony output of specific cells in each of our single cell differentiation assays. **We believe this in an important distinction and we have carefully proofread the revised manuscript so that there can be no confusion in this wording.**

We concur that the most important finding is that “Cells at the CLEC9A^{hi}CD34^{lo} extreme contained long-term repopulating multipotent HSCs”, as we had originally stated in the abstract. Some may perceive an apparent contradiction between the fact that most cells within Subset1 produce My/Ery colonies in our *in vitro* assays (verified in 3 independent assays) and the fact that most mice transplanted with low doses of Subset1 (but not single cells) display multipotent engraftment. However, we believe that these 2 findings can be reconciled taking into account the points below.

- 1) The *in vivo* assays are performed at limiting dilution (LDA) which means that engraftment is likely generated by only a few cells, which are selected by the assay based on their self-renewal capacity. One limitation of the *in vivo* assay is that it effectively only informs on the lineage potential of the cells that have self-renewal capacity. In contrast, the *in vitro* assays assess the lineage potential of all cells within that population independently of their self-renewal potential.
- 2) In our *in vitro* single cell assays, we find that approximately 2% of single CD49f+ Subset1 cells produce My/Ery/Meg/NK or My/Ery/NK colonies. Our *in vivo* LDA estimates 1 in 13 cells (7%) to be long-term repopulating and capable of giving rise to Ly cells. Given the limitations and uncertainties in the measure associated with each assay, it is possible (even though this cannot be formally proven) that only the cells that read-out multipotent *in vitro*, also repopulate long-term and produce Ly cells *in vivo*. In any case, based on both assays, it is extremely likely that multipotent long-term repopulating cells are rare within the phenotypic 49f+ Subset1 compartment.
- 3) Another possibility is, as the reviewer suggests, that Ery differentiation proceeds faster than Ly differentiation *in vitro*. Importantly, in the assay where we can simultaneously test Ery and NK (Ly)

output, both types of cells were efficiently generated within 2 weeks of culture and continuing the culture longer did not increase the percentage of colonies reading out Ly (data not shown). Importantly, as the xenograft model sustains robust production of B cells but not of NK cells, we had also used a single cell assay where My, B cell and NK cells can be simultaneously differentiated (Figure 2h). This assay is recognised as the gold-standard assay to assess B cell production in human stem and progenitor cells and has been used widely by many groups (Doulatov et al., Nat. Immun., 2010, Laurenti et al., Nat Immun, 2013, Görgens et al., Cell Reports, 2013, Van Galen et al., Cell Stem Cell, 2014, Karamitros et al., Nat Immun., 2018). Additionally, in this assay, the majority of single cells from Subset1 cannot give rise to either B or NK cells, but about 5-10% do.

- 4) Given the body of literature on the heterogeneous behaviour of mouse and human phenotypic HSCs, it would be naive to think of Subset1 or Subset2 as homogeneous populations. We clearly describe in Figure 2d-e that these populations are enriched in cells with particular differentiation behaviours but are not pure and were careful in our description not to convey such an idea. In addition, the data presented in Figure 4 from the original manuscript shows that within 5 days of differentiation *in vitro*, cells from 49f⁺ Subset1 can either i) retain the same differentiation capacities as the original population; ii) generate single cells that now can only produce My, My/Ery or Ery colonies; iii) generate single cells that can only give rise to My, My/Ly or Ly colonies. We also estimated that during these 5 days single cells make 2 to 3 divisions. This shows that there are distinct trajectories of differentiation from distinct Subset1 cells, supporting the idea that cells with different lineage properties coexist within Subset1, a small proportion (estimated 5-15%) of which can make Ly.

We believe that the most likely interpretation of our data is that Subset1 contains i) rare multipotent self-renewing stem cells (which sustain the *in vivo* engraftment, and possibly may read out multipotent *in vitro* as well), ii) other cells that do not self-renew, and hence do not read out *in vivo*, the identity of which we can only speculate based on *in vitro* assays. Putting together all the data from our 4 distinct *in vitro* assays, we propose that the majority of single cells in Subset1 are non self-renewing and either restricted or heavily biased towards My and Ery fates.

In our original manuscript, we had stated in the discussion: “Within the CLEC9A^{hi}CD34^{lo} fraction of the HSC/MPP and 49f⁺ HSC pools (Subset1), multipotent (Ery/My/Ly) long-term repopulating HSC coexist with cells with restricted My/Ery potential (as read out *in vitro*)”. **Based on the reviewer’s comment, we have nonetheless made several changes in the manuscript so that it is very clear that:**

- i) **Multipotent cells within Subset1 are rare (now on line 469 and 516);**
- ii) **The statement that other cells within Subset1 may be either restricted or heavily biased towards My and Ery fates is at this current stage no more than an hypothesis (now on line 518).**

As also requested by Reviewer 2, we have also added a schematic diagram of the most likely model of lineage differentiation supported by our findings (Figure 6h).

4: While it is appreciated that some level of simplification can facilitate manuscript writing I think referring to Meg-E, myeloid and lymphoid cell types as three lineages is misleading, and in particular referring to clones with e.g. myeloid cells only as unilineage, is unwarranted, since there are multiple myeloid cell lineages. It should not be difficult to identify monocytes and neutrophils by FACS with the antibody panel used and describe the readouts accordingly.

During our initial phenotyping of the colonies grown in My/Ery/Meg/Ly conditions, we had included antibodies against CD41 (for megakaryocytes) but also CD14 and CD15, respectively marking Monocytes and Granulocytes (neutrophils). We have thus reanalysed our data treating Meg and Ery outputs as 2 distinct lineages as well as separating My outputs in Monocytic (Mono; CD14⁺), Granulocytic (Gran; CD15⁺), Monocytic and Granulocytic (MonoGran; CD14⁺ cells together with CD15⁺ cells) and immature My cells (Undiff; CD11b⁺ but CD14⁻CD15⁻). This results in 32 different types of colonies and found no new finding from this classification. As this reviewer rightly appreciates, using all of these makes it difficult to highlight the important messages to the reader. So, in response to comments from both this Reviewer and Reviewer 2, we now present the data in the following manner:

- 1) **Our analysis now treats the Ery and Meg as distinct lineages in all single cell differentiation assays.** We thus describe 10 possible types of colonies, of 4 distinct lineages, arising from the HSC/MPP pool or the 49f⁺ HSC compartment (see table below). We also observed an 11th type of colony, MyMeg, which was only produced by Diff1 cells (Figure 4c, CLEC9A^{hi}CD34^{lo} cells originating from 49f⁺ Subset1 after 5 days in differentiation conditions).

Colony Type	Classification
My/NK/Ery/Meg	Quadrilineage
My/NK/Ery	Multilineage
My/Ery/Meg	

My/NK	
My/Ery	
My/Meg	
Ery/Meg	
My	Unilineage
NK	
Ery	
Meg	

This classification is now used in the new Figures 1a, 1c, 2d-e and 4c, and the text has been changed accordingly. Importantly the conclusions of our analysis remain unaltered. This is because, as we had pointed out in the original manuscript, there are few unilineage Meg colonies arising from HSC/MPP pool, 49f⁺ HSCs or their subsets in this assay (<1% of all colonies analysed). There are a more multilineage colonies containing Meg and another lineage (up to 23% from Subset1), but with the exception of colonies derived from populations cultured for 5 days before being plated in this assay (Figure 4c), these are always associated with the Ery lineage.

- 2) In the analysis we present in the revised manuscript, My corresponds to Mono, Gran, and/or MonoGran outputs. But to provide a complete picture, **we now also separately show the distribution of My sublineages for all My colonies obtained from each population (Supplementary Fig. 1e, 3c-d and Suppl. Figure 5c of the revised manuscript).** We observe that 49f⁺ Subset2 single cells produce significantly less MonoGran colonies than 49f⁺ Subset1, and that both Subset2 and 49f⁺ Subset2 single generate significantly more Mono colonies than Subset1 and 49f⁺ Subset1 respectively.

We thank the Reviewer for this helpful suggestion as we believe this new analysis makes our manuscript more comprehensive, and thus more useful to the larger community studying different lineages.

Reviewer #2 (HSC, differentiation)(Remarks to the Author):

This is an elegant and well written paper describing a detailed analyses of human haematopoietic stem and progenitor cells. The study combines single cell index sorting, the analyses of differentiation capacity of sorted cells and single cell RNA sequencing to analyse the transcriptome of these cells. Through a series of elegant and well-presented experiments this paper identifies several novel findings that make this paper worthy of publication in Nature Communication.

It is certainly interesting to the haematopoiesis community but the approach and the analyses they have done would be relevant to other fields

Key findings:

- They identify a novel marker combination for HSPCs (defined by reciprocal levels of expression of the markers CD34 and CLEC9) using single cell sorting, functional analysis and single cell sequencing. They demonstrate that cells sorted based on the use of this new marker combination defines specific HSC 49f subpopulations with distinct functional outputs both in vitro and in vivo. Single cell sequencing defines the transcriptional profile of these cells. They define the hierarchical organization of two subsets with 49f⁺ Subset1 (CLEC9Ahi CD34lo) cells being able to produce cells with a phenotype equivalent to that of 49f⁺ Subset2 (CLEC9Alo CD34hi but 49f subset 2 cells were unable to give rise to 49f subset 1.
- The study describes a novel transplantation to identify progenitors with erythroid potential by injection of human EPO into transplanted immunodeficient mice. This allows identification of cell populations that were not able to be analysed in vivo previously. This in vivo model defines the functional output of their HSPC subsets, defined by CD34/CLEC9 expression.

• Most importantly the study provides convincing evidence that supports the proposal that lineage commitment occurs within the HSC pool rather than downstream of this this population as assumed previously. This adds to the growing body of evidence in the field that the haematopoietic hierarchy is more complex than the classical model.

The data they present supports their conclusions with appropriate statistical analysis performed on all the data.

We thank the Reviewer for qualifying our work as elegant, well-presented and providing important novel insights to the field.

Specific comments.

1. One caveat to their study is that the cells are cultured in vitro for a number of days prior to analyses and thus it is formally possible that their findings represents events within this culture model. They should formally state this as one of the limitations as it may not precisely recapitulate events in vivo. Nevertheless their findings will certainly inform attempts to generate blood cell lineages in vitro from sources such as iPSCs.

We fully agree with the Reviewer that this limitation was not pointed out explicitly enough in our first manuscript. **We have now added this on line 475 in the discussion.**

2. Lines 103-114 and Figure 1a,c. I am confused by the authors definition of unilineage and really don't understand why they have stated they include EryMeg (clearly bi-lineage) in that subset. Some clarification is required. The following statements seems contradictory-
"We therefore considered EryMeg colonies as unilineage colonies and focused on the three main lineages: My, Ly and Ery (including EryMeg). Only 3.5% of single HSC/MPP pool cells produced multilineage colonies comprising all three lineages (Fig. 1a). Approximately 50% of 110 the cells produced bilineage colonies, predominantly of the My/Ery type"

Pooling Ery and Meg fates into one lineage was confusing to both this Reviewer and Reviewer 1. As pointed out in response to the third point of Reviewer 1, we have now re-analysed our data treating these 2 lineages separately. **The text and figures have been updated as described above to reflect this new analysis.** This produces results consistent with our previous messages but we believe significantly clarifies our manuscript. We thank both reviewers for this suggestion.

3. Figure 1e – clarification on how the data has been manipulated to generate this figure. Does each point represent the mean time for cells from a single cord? (n=4 but there are only 3 points).

We have indeed measured time to first division in 4 independent experiments (each with a different cord blood sample) in Figure 1e. But we agree with the Reviewer that this was not evident in the original Figure 1e, because some of the measurements are very close. **We have now improved the visibility of these 4 points. In addition the word "mean" has been added to the y axis label of this figure. Similar changes have been made to Figure 2g.**

We had originally stated in the figure legend that each point represents the "Mean time to first division", which is calculated as the EC50 of non-linear fit of cumulative first division kinetics. We recognise that this may not be clear enough for a reader who is not very familiar with these assays. This is calculated by first measuring the time of first division for each single cell in each experiment, plotting and fitting a cumulative curve similar to that in Figure 2f, and then estimating the point of the curve at which 50% of the cells have divided (EC50). **We have now added the cumulative curves relative to Figure 1e in Supplementary Fig. 1h. In addition we have specified the number of single cells used for each of the 4 individual experiments indicated.**

4. The manuscript would benefit from having a schematic 'haematopoietic hierarchy' figure that summarises their findings.

We thank the reviewer for this suggestion that helps highlighting the main messages of our work to the reader. **We have now added a schematic in Figure 6h** which summarises the following important findings:

1) The transcriptional and functional continuum observed at single cell level along the anti-correlating CLEC9A/CD34 axis.

2) The concept that Subset1 contains rare multipotent and long-term repopulating HSCs, but also many single cells with no self-renewal, the nature of which remains to be fully understood, but which based on our data we speculate to be either restricted or heavily biased towards the Ery and My lineages.

3) The isolation of a My-Ly restricted Ery-null cell type with infrequent but durable repopulation capacity (Subset2), upstream of, and functionally and molecularly distinct from LMPPs.

Reviewer #3 (System biology, RNA-seq)(Remarks to the Author):

The authors analysed single cells from human haematopoietic stem cell (HSC) subsets with differentiation assays and single-cell RNA-seq. They observed that CLEC9A and CD34 cell surface expression correlates with lineage potential and HSC transcriptome profiles. They provide evidence that first lineage restriction events occur already within the HSC compartment to generate myelo-lymphoid-committed cells with no erythroid differentiation capacity.

Up to 8 cell surface markers have been used to distinguish HSC subsets with predetermined functional polarisation. It is an interesting study, but as I'm not a researcher in this particular field, I can not fully appraise the relevance. The textbook model of haematopoiesis involves a series of discrete cell stages. At a closer (single-cell) look, this is rather a continuous process. In this regard, this study complements novel findings published by others, and it is also likely that future studies will identify additional markers to further delineate in even more detail when and how distinct lineages emerge.

Belluschi et al. used appropriate technologies and tools for data analysis. The manuscript is well written and includes recent literature. The conclusions seem to be supported by the data. I have few comments that may help to improve the paper.

We thank the reviewer for their positive assessment of our manuscript, and all the useful comments that we have addressed below.

1. In several assays, the HSC/MPP pool and the 49f+ HSC subset showed similar outcomes. I wonder whether this was previously anticipated by the authors. They may better emphasize the differences (to readers not familiar with previous papers), e.g., as those seen in Suppl Table 3.

The Reviewer is right in pointing out that the outcomes of the analysis of different subsets isolated from either HSC/MPP pool or 49f+ HSC are overall relatively similar. However generally, Subset1 and Subset2 isolated from 49f+ HSC display characteristics of more immature cells than the corresponding subsets isolated from HSC/MPP pool. This is based on i) highest frequency of long-term repopulating cells for 49f+ Subset1 (Suppl. Table 3); ii) increased transcriptional lineage priming observed in both HSC/MPP pool derived Subset1 and Subset2 compared respectively to 49f+ Subset1 and 49f+ Subset2 (Fig.3f-g, Fig.6c).

We agree with the Reviewer that this is worth pointing out to the reader, and **have thus now added a sentence in the discussion to highlight the more immature nature of 49f+ HSC-derived subsets (line 466).**

2. Abstract [page 1, line 17]: add CD19-

The definition of 49f+ HSC has now been corrected in the abstract.

3. Results: a) [page 5]: add percentages of 49f+ and 49f- cells in HSCs (only mentioned in the Suppl).

We have now added in the main text the percentages of phenotypic 49f+ and 49f- cells found within the HSC/MPP pool (approximately 13%, line 98). We have also added on line 350 the mean percentages of phenotypic 49f+ HSCs within Subset1 and Subset2.

b) [page 5, line 94] why was CD117 included?

Levels of cell surface CD117 have been shown previously to be heterogeneous across the human HSC/MPP pool and to correlate with distinct repopulation capacities of HSCs. As such we thought this marker was worth tracking in our experiments. We agree with the Reviewer that this should be specified in the text and **have thus added this explanation and the accompanying reference (Cosgun et al., Cell Stem Cell, 2014) on line 95.**

c) 49f+: the second marker CD90 could be mentioned more often.

The definition of 49f+ HSC is well accepted in the field and indeed is based on the use of a second marker, CD90. To facilitate readability, we had defined what we refer to as 49f+ HSC (which are CD90+) and 49f- HSC (which are CD90-) in the first paragraph of the results and in Supplementary Figure 1a. This definition is then used throughout the paper. **We agree with the Reviewer that it may be useful to remind the reader that the original definition of this population includes CD90, and thus we have added this in the intro (line 64).**

4. Fig 1d: I assume that the boxes are wrongly labelled (My/Ery => My/NK) – please check. Moreover, there are two identical significance lines below the boxplot for PC2. Please check whether the numbers given in the legend [page 40, line 914 and line 917] and in the Results [page 5, line 91] are correct: “435 cells/colonies from 6/5 CB samples.”

We apologise for these inaccuracies and thank the Reviewer for spotting them. The My/Ery and My/NK labels in Figure 1d were indeed swapped, **this has now been corrected.** In addition, the bottom significance line for the PC2 axis was redundant as the only statistical difference is observed between My/Ery and My/NK colonies on PC2. **This redundant line has now been eliminated.**

We also carefully checked all the numbers given in the figure legend, which were correct. However there were inaccuracies in the main text. Line 92 in the main text should have said 6 CBs (and not 5) and line 131 should have read 819 49f+ HSC from 4 individual CBs (and not 900). **These numbers have now been corrected.**

5. Fig 5f / 5j: n=2 for Subset2 is a bit critical. If possible, increase sample size to provide more confidence in the results. Otherwise, justify why only 2 mice were used.

In our original manuscript, there were indeed only 2 mice assessed in Figure 5f and 5j. These plots show the lineage composition of the graft found in mice transplanted respectively with Subset2 cells (8 week analysis) and 49f+ Subset2 cells (20 weeks analysis). Importantly for these experiments, we initially transplanted respectively 8 and 13 mice (Figure 5c and 5i). Because of the low frequency of engrafting cells in Subset2, only 2 of these mice engrafted in each case. This is not unusual for LDA experiments and explains why there are only 2 mice shown for Subset2 and 49f+ Subset2 in Figure 5f and 5j.

For this round of revision, we have now included data from an additional repeat of a 20 week transplantation experiment from 49f+ Subset1 and 49f+ Subset2 cells. For this experiment we have transplanted an additional 9 mice with 49f+ Subset1 cells and 8 mice with 49f+ Subset2. **The results from these additional animals are now shown in Figure 5i, and this data has been included in an updated calculation of the frequency of repopulating cells (revised Figure 5h and Supplementary Table 4)** Of the 8 mice transplanted with 49f+ Subset2, 1 successfully engrafted. **Figure 5j now shows n=3 for 49f+ Subset2. The new numbers of animals**

has been updated in the Figure legend, which now also contains the number of transplanted animals for panels b,c,d and i.

6. Suppl Fig 2a (right one): I assume that the boxes are wrongly labelled (My/Ery \leftrightarrow My/NK).

We apologise for this switch in the labels. **This has been corrected in Supplementary Figure 2a.**

7. Suppl Table 2: CD34 and CLEC9A are not in this table, which is not intuitive – please discuss and report fold-changes for these genes.

CD34 and CLEC9A are not in Supplementary Table 2 because this table only shows genes differentially expressed between single 49f⁺ Subset1 and 49f⁺ Subset2 cells. In this comparison, CD34 and CLEC9A have significant nominal p-values (CD34: p=0.033, fold-change= 1.47 fold higher in 49f⁺ Subset2; CLEC9A: p= 0.039, fold-change=1.9 fold higher in 49f⁺ Subset1) but both have FDR > 0.05. For the Reviewer's information we present below violin plots for the normalised expression of these genes, as well as boxplots of the cell surface expression of the single cells sorted for this sc-RNA-seq experiment. As can be seen below, levels of CD34 and CLEC9A at the surface confirm that the flow cytometry sorting for this experiment was correct. We attribute the relatively low fold-changes at the RNA level to the low sensitivity/high variance of sc-RNA-seq. Indeed, when we performed 20 cell RNA-seq on Subset1 and Subset2 cells, which has much higher sensitivity/less variance, we then found that CD34 mRNA levels were significantly higher by DESeq2 in Subset2 compared to Subset1 (FDR=2.45x10⁻⁶; fold-change= 1.92), and that CLEC9A levels were significantly higher in Subset1 compared to Subset2 (FDR= 4.72x10⁻⁶; fold-change= 5.67).

Figure 2: a, Log₁₀ normalised expression of CLEC9A (left) and CD34 (right) from 78 49f⁺ Subset1 single cells and 75 49f⁺ Subset2 single cells as measured by single cell RNA-seq. **b,** Cell surface expression of CLEC9A (left) and CD34 (right) for the same single cells shown in a as measured by index sorting.

For completeness, we are now presenting the table of differentially expressed genes for Subset1 and Subset2 as Supplementary Table 3. We also added the fold-change values for CD34 and CLEC9A in the table legends for both Supplementary Table 2 and 3.

8. Suppl Table 3: What was the statistical test to compare repopulating capacity?

We thank the Reviewer for spotting that the test used was only written in the legend of Figure 5, but not in that of the original Supplementary Table 3 (now Supplementary Table 4 in the revised manuscript). **We have now added in the legend that the p-value in the revised Supplementary Table 4 is calculated using the ELDA method (Hu and Smith, J. Immunol. Methods, 2009).**

9. Discussion:

a) [page 18, line 406-8]: Is it really justified to write “whole” and “unipotentiality”? Rephrase

According to the Reviewer's suggestion, we have rephrased as follows: “Our study provides direct evidence that the developmental transition from true multipotentiality to bipotentiality occurs already within the phenotypic HSC/MPP compartment.” (line 456).

b) [page 20, line 463]: define abbreviations MPP2 and CMRPs. Spell out

MPP was already defined as Multipotent Progenitor in the introduction (line 36). We have now spelled out Common Myeloid Repopulating Progenitors (CMRPs) in the discussion on line 520.

c) The authors focused on extreme cell populations. However, they may also speculate (more directly) about CD49f+CD90- and CD49f-CD90+ cells and CLEC9AloCD34lo and CLEC9AhiCD34hi cells, which represent fairly large fractions.

We now state on line 472: “. Our data predict that cells with intermediate cell surface phenotypes (such as CLEC9A^{lo}CD34^{lo} and CLEC9A^{hi}CD34^{hi}) display heterogeneous behaviours that will need to be further explored.”

CD49f⁺CD90⁻ and CD49f⁺CD90⁺ cells have been described to limited extent in Notta et al., Science, 2011.

d) The authors should discuss transcription factors in more detail (e.g., those identified by GSEA analysis and those reported in Suppl Tables 1 and 2).

We have now mentioned in the discussion some important transcription factors that have been identified by our ICGS and GSEA analyses. **We now cite MAFF, JUND, GATA2, HOXA9 and HLF on line 495.**

Overall, we thank this Reviewer for their comments on how to improve our discussion. Of note, we have kept our modifications short to keep in line with Nature Communications manuscript length restrictions.

10. Methods: Please proofread and revise some minor typos. Check plural forms, consistency in terminology, definition of relevant abbreviations, formatting.

We have now proofread the methods section carefully and corrected inconsistencies in terminology and formatting.

11. The single-cell RNA-seq data have been deposited in GEO. However, the data are still private (no token provided). I could not check whether the experimental methods are described in sufficient detail in these GEO records. The authors should add the 20-cell RNA-seq data as GEO SubSeries.

We sincerely apologise for a miscommunication with the editor that has resulted in GEO token not being transferred to the Reviewers. Indeed when we submitted our manuscript the GEO access token was mentioned on our comments to the editor on the online form (and not in the manuscript) as requested by Nature journals. The following tokens (weblinks in methods) can be used to access all the RNA-seq data presented in the revised manuscript:

- 1) Superseries GSE115798; token: ovirmcierbmzlmx
- 2) Series GSE104995 – for single cell RNA-seq of 49f⁺ HSC, 49f⁺ Subset1, 49f⁺ Subset2 and 20 cell RNA-seq of Subset1 and Subset2; token: ovirmcierbmzlmx
- 3) Series GSE115639 - for the novel dataset of single cell RNA-seq comparing 49f⁺ Subset2, Subset2, LMPPs and MLPs provided in this revised manuscript; token: ivotysoivvsdpwv

Reviewers' comments:

Reviewer #1 (Remarks to the Author):

In response to the comments to the initial submission the authors have added new data and several modifications to the text. Overall, this has significantly improved the manuscript, and in particular the inclusion of new data on LMPPs is appreciated. There are, however, still some issues, mainly of a conceptual nature, that it would be appropriate to address.

1. One of the pitfalls of single cell biology is that the sheer number of data points generated statistical significance where biological significance is not necessarily obvious. This can lead to statements that can be misleading. For example (p8):

"In addition, among 49f+ HSCs that generated My/NK colonies, those cells that gave rise to My/NK colonies with a Ly- bias (number of NK cells in the colony greater than number of My cells) had significantly lower levels of CLEC9A and significantly higher levels of CD34 at the time of sort than cells giving rise to My-biased My/NK colonies (Supplementary Fig. 2b). Similarly, cells generating My/Ery colonies with an Ery-bias (number of Ery cells in the colony greater than number of My cells) had significantly lower CD34 expression than those producing My-biased My/Ery colonies (Supplementary Fig. 2b)."

Here (and elsewhere where similar statements are made) should be made clear that this applies to average expression, as it is clearly not true that single cells within the populations display these differences, which are fairly small compared to the variance. That such a small effect size is statistically detectable is, as discussed, mainly due to the very large sample size. It would be useful if the actual average fold-differences could be made clear, as this would help the reader evaluate the usefulness of the markers for practical purposes.

2. For the second point raised the authors use FACS purity to address the issue of whether the repopulating cells in Subset2 are a contaminant. The point here is whether the cells that provide the repopulating activity are representative of the population: if a small subset of Subset2 cells are repopulating (which seems likely give the cell number required for engraftment), and the majority of the cells not, the differentiation properties of the overall population will not necessarily reflect those of the repopulating subset. The only way to resolve this would be to measure repopulating activity at single cell resolution, which is impractical, or to separate the repopulating cells from the bulk, which may not be possible. It is therefore not reasonable to expect this to be resolved experimentally at present. However, it would be appropriate to state that repopulating activity is not necessarily associated with the molecular and cellular properties of the overall population.

3. Regarding the issue of whether low engraftment (and therefore possibly lack/loss of long-term repopulating cells) is associated with the Ly-bias, it seems to me that the graphs provided by the authors actually support this notion: My and Ery engraftment is clearly associated with high overall engraftment (especially if total, as opposed to relative, engraftment is considered). Here actual statistical analysis would be appropriate (and maybe a Supplementary Table with the observed values). This would include testing whether the lack of Ery output from Subset2 cells is statistically significant.

4. The use of the term "My/Ery restricted" implies that the fate of the cell(s) is pre-determined. However, given the stochastic nature of lineage readout it is not possible to reach this conclusion by analyzing the output of a single cell: it cannot be ruled out that other lineage potentials were present, but not detected. This is in particular true if the population from which the single cell is derived has additional potentials (if a potential is absent across the entire population it would be statistically

appropriate to state that the single cells are fate-restricted, but this is not what is observed here). A statement of My/Ery fate restriction within the Subset1 therefore cannot be supported by the current data. While the scenarios put forward are plausible, it is equally possible that the entire cell population contains Ly-potential, which under the assay conditions is read out in 10% of the cells. This needs to be objectively discussed.

Reviewer #2 (Remarks to the Author):

All points that I raised in my first review (reviewer 2) have been addressed.

Reviewer #3 (Remarks to the Author):

The authors addressed my comments and improved the manuscript accordingly. I recommend the manuscript for publication. A few typos remain to be corrected, e.g.

line 452: "potential occur" => "potential, occur"

line 819: "populations studies" => "populations studied"

Please find below our response to the Reviewers' comments on our manuscript. We are very pleased to see that they all consider our manuscript to have significantly improved during the last round of revision. As you will appreciate below, we have now addressed all remaining concerns by providing statistical analyses, the raw data relative to Supplementary Figure 6c (Figure 1 of our previous rebuttal letter), clarifying the limitations of our assays, their interpretation and correcting typos.

We are grateful to the Reviewers for their additional constructive comments.

Reviewer's comments are in blue, our detailed response in black, and highlighted in bold are the changes we have made to the manuscript.

Reviewers' comments:

Reviewer #1 (Remarks to the Author):

In response to the comments to the initial submission the authors have added new data and several modifications to the text. Overall, this has significantly improved the manuscript, and in particular the inclusion of new data on LMPPs is appreciated. There are, however, still some issues, mainly of a conceptual nature, that it would be appropriate to address.

We are pleased to hear that the Reviewer concurs that our previous modifications have significantly improved the manuscript.

1. One of the pitfalls of single cell biology is that the sheer number of data points generated statistical significance where biological significance is not necessarily obvious. This can lead to statements that can be misleading. For example (p8):

"In addition, among 49f+ HSCs that generated My/NK colonies, those cells that gave rise to My/NK colonies with a Ly- bias (number of NK cells in the colony greater than number of My cells) had significantly lower levels of CLEC9A and significantly higher levels of CD34 at the time of sort than cells giving rise to My-biased My/NK colonies (Supplementary Fig. 2b). Similarly, cells generating My/Ery colonies with an Ery-bias (number of Ery cells in the colony greater than number of My cells) had significantly lower CD34 expression than those producing My-biased My/Ery colonies (Supplementary Fig. 2b)."

Here (and elsewhere where similar statements are made) should be made clear that this applies to average expression, as it is clearly not true that single cells within the populations display these differences, which are fairly small compared to the variance. That such a small effect size is statistically detectable is, as discussed, mainly due to the very large sample size. It would be useful if the actual average fold-differences could be made clear, as this would help the reader evaluate the usefulness of the markers for practical purposes.

We understand the Reviewer's concern that single cell biology studies bring perspectives and

statistical analysis challenges distinct from those of more classical cell population studies. Thus the biological relevance of some findings may need clarification and it is helpful to give the reader the amplitude of the effects observed and explain their relevance.

We have thus made the following changes (highlighted in bold) in the text:

- 1) On line 158: “Consistently, single cells that produced Ery or My/Ery colonies had significantly lower levels of CD34 and significantly higher levels of CLEC9A cell surface expression than those producing My/NK or NK colonies (**Fig. 1f and Supplementary Fig. 2a, median fluorescence intensity shifted respectively by 40% for CLEC9A and 13% for CD34**). **Interestingly, milder but significant shifts in expression of CLEC9A and CD34 cell surface expression were also observed between 49f⁺ HSCs that generated My/NK colonies with a Ly-bias (number of NK cells in the colony greater than number of My cells) and those with a My-bias, as well as between cells generating My/Ery colonies with an Ery-bias (number of Ery cells in the colony greater than number of My cells) and those with a My-bias (median fluorescence intensity shifts of 4 to 25%, Supplementary Fig. 2b)**. **These data indicate that cells with distinct differentiation outputs are distributed on a continuum of anticorrelating CLEC9A and CD34 cell surface expression.**”
- 2) **The amplitude of the median fluorescence intensity shifts has also been added to the figure legend of Supplementary Fig.2b**
- 3) On line 146 and 219: we have added “on average”

Regarding changes 1 and 2, it is worth noting that because flow cytometry measurements are not distributed normally, we have chosen to provide an estimation of the shift in median fluorescence intensity seen in distinct populations, as this is a more appropriate measure than the average and/or fold-changes. Also the values shown in Supplementary Fig.2b are logicle transformed and normalised (as described in methods so that independent experiments could be appropriately compared) and do not directly reflect the amplitude of changes seen in the raw flow cytometry measurement, but rather underestimate it. Here we have calculated the shifts in median fluorescence intensity between the different conditions that best give a sense of the relevance of the shifts that can be expected from these flow cytometry markers.

Finally, in terms of relevance of these shifts, the functional data shown in Figure 2, 4 and 5 demonstrates that 40% shift for CLEC9A and 13% shift for CD34 are sufficient to allow significant enrichment of cells with distinct differentiation behaviours. The shifts observed between cells that give rise to My/Ly colonies *in vitro* with either more My cells (My-biased) or NK cell (Ly-biased) are milder, as we now state in the text. Such shifts are not sufficient to prospectively purify these populations. Nonetheless, we believe this observation to be relevant as it contributes to illustrate to the reader the concept of “continuum” of differentiation capacity along a gradient of cell surface markers. The latter concept has been described in other contexts by other published studies with effect sizes of the same magnitudes as in our study (Velten et al, 2017; Karamitros et al., 2018, Knapp et al., 2018).

2. For the second point raised the authors use FACS purity to address the issue of whether the repopulating cells in Subset2 are a contaminant. The point here is whether the cells that provide the repopulating activity are representative of the population: if a small subset of Subset2 cells

are repopulating (which seems likely give the cell number required for engraftment), and the majority of the cells not, the differentiation properties of the overall population will not necessarily reflect those of the repopulating subset. The only way to resolve this would be to measure repopulating activity at single cell resolution, which is impractical, or to separate the repopulating cells from the bulk, which may not be possible. It is therefore not reasonable to expect this to be resolved experimentally at present. However, it would be appropriate to state that repopulating activity is not necessarily associated with the molecular and cellular properties of the overall population.

The reviewer is right to highlight this issue that is not restricted to our work but common to all studies pertaining to human HSCs: transplantation in xenografts only reads out a minority of cells as repopulating (which could be an underestimation because of the limitations of the model), whereas single cell molecular assays will provide measurements for each cell in a particular population. It is obviously not possible to assay the same cell both at the molecular level and in a functional assay and if a particular functional behaviour is rare in a population, the average characteristics of this population may not apply to these rare cells. This may not be clear to all readers, so, as suggested by the reviewer, we have specified in the discussion on line 540 that **“the rare Subset2 cells with long-term repopulation capacity may not share the same molecular properties of the overall Subset2 population”**.

3. Regarding the issue of whether low engraftment (and therefore possibly lack/loss of long-term repopulating cells) is associated with the Ly-bias, it seems to me that the graphs provided by the authors actually support this notion: My and Ery engraftment is clearly associated with high overall engraftment (especially if total, as opposed to relative, engraftment is considered). Here actual statistical analysis would be appropriate (and maybe a Supplementary Table with the observed values). This would include testing whether the lack of Ery output from Subset2 cells is statistically significant.

We agree with the Reviewer that this is an important point for our manuscript. Indeed, in our original revision, **we had already shown the Figure1 from the response to the reviewers as Supplementary Figure 6c, and had provided the relevant statistics both in the main text (line 374) and in the figure legend**. As specifically requested by the Reviewer, we had already tested that the lack of Ery output from Subset2 cells is statistically significant compared to Subset1. This is the case considering all animals ($p=0.0149$), but also restricting our analysis to animals with higher overall engraftment to exclude any potential confounding effects of the size of the graft and its lineage composition ($p=0.012$). This analysis confirms that the absence of Ery cell production from Subset2 is a robust finding.

For completeness, in this revision, we have added all the observed values used in Supplementary Figure 6c as a new Supplementary Table 5 (line 376).

4. The use of the term “My/Ery restricted” implies that the fate of the cell(s) is pre-determined. However, given the stochastic nature of lineage readout it is not possible to reach this conclusion by analyzing the output of a single cell: it cannot be ruled out that other lineage potentials were present, but not detected. This is in particular true if the population from which the single cell is derived has additional potentials (if a potential is absent across the entire population it would be

statistically appropriate to state that the single cells are fate-restricted, but this is not what is observed here). A statement of My/Ery fate restriction within the Subset1 therefore cannot be supported by the current data. While the scenarios put forward are plausible, it is equally possible that the entire cell population contains Ly-potential, which under the assay conditions is read out in 10% of the cells. This needs to be objectively discussed.

We thank the Reviewer for their remark that helped us make the manuscript clearer. Indeed on line 520, we now use the word “hypothesise” instead of “propose” (“We **hypothesise** that these coexist with cells with restricted My/Ery potential (Figure 6h; **as suggested by experiments *in vitro* and *in vivo* at 2 weeks post-transplantation**, which would share similarities with mouse MPP2^{29,30} and Common Myeloid Repopulating Progenitors²⁸”). In addition, we now specifically describe the limitations of the assays that preclude a formal conclusion and state that future experiments would be needed to prove or disprove this hypothesis: “**Nonetheless, as we cannot formally rule out that our assays underestimate the number of single cells with Ly potential in Subset1, future studies will have to address if truly My/Ery restricted or largely My/Ery biased single cell behaviours exist in this compartment.**” (line 523).

Reviewer #2 (Remarks to the Author):

All points that I raised in my first review (reviewer 2) have been addressed.

Reviewer #3 (Remarks to the Author):

The authors addressed my comments and improved the manuscript accordingly. I recommend the manuscript for publication. A few typos remain to be corrected, e.g.

line 452: "potential occur" => "potential, occur"

line 819: "populations studies" => "populations studied"

We have now corrected these typos and others after careful proofreading of the manuscript.

REVIEWERS' COMMENTS:

Reviewer #1 (Remarks to the Author):

The authors have now addressed all remaining concerns satisfactorily. I am happy to support publication at this point.

REVIEWERS' COMMENTS:

Reviewer #1 (Remarks to the Author):

The authors have now addressed all remaining concerns satisfactorily. I am happy to support publication at this point.

We are pleased to hear that Reviewer #1 is now also in favour of publication of our manuscript and thank him once more for his very constructive feed-back and suggestions.